# Prediction of a topological $p + ip$ excitonic insulator with parity anomaly

Rui Wang[1,2], Onur Erten[3], Baigeng Wang[1] & D.Y. Xing[1]

Excitonic insulators are insulating states formed by the coherent condensation of electron and hole pairs into BCS-like states. Isotropic spatial wave functions are commonly considered for excitonic condensates since the attractive interaction among the electrons and the holes in semiconductors usually leads to $s$-wave excitons. Here, we propose a new type of excitonic insulator that exhibits order parameter with $p + ip$ symmetry and is characterized by a chiral Chern number $C_c = 1/2$. This state displays the parity anomaly, which results in two novel topological properties: fractionalized excitations with $e/2$ charge at defects and a spontaneous in-plane magnetization. The topological insulator surface state is a promising platform to realize the topological excitonic insulator. With the spin-momentum locking, the interband optical pumping can renormalize the surface electrons and drive the system towards the proposed $p + ip$ instability.

[1] National Laboratory of Solid State Microstructures, Collaborative Innovation Center of Advanced Microstructures, and Department of Physics, Nanjing University, Nanjing 210093, China. [2] Department of Physics and Astronomy, Shanghai Jiao Tong University, Shanghai 200240, China. [3] Department of Physics, Arizona State University, Tempe, AZ 85257, USA. Correspondence and requests for materials should be addressed to O.E. (email: onur.erten@asu.edu) or to B.W. (email: bgwang@nju.edu.cn)

Massless Dirac particles preserve both parity and time-reversal symmetry (TRS). However, an infinitesimal mass will inevitability break these symmetries through the radiative correction[1–3], resulting in an Abelian or nonAbelian Chern–Simons theory of the external gauge field. This effect is called parity anomaly. Quasiparticles with linear dispersion in condensed matter physics have recently drawn broad attention[4–13], and they provide a quintessential platform to test these effects. Indeed, the gapless Dirac fermions of topological insulator (TI) surface states have been predicted to exhibit the parity anomaly and a number of exotic phenomena when different types of mass terms are considered, including the $p + ip$ type topological superconductor induced by the superconductor (SC) proximity effect[14,15], and the quantum anomalous Hall effect in the TI surface with a Zeeman field[16–19].

In this work, we expand the realm of two-dimensional (2D) topological states by devising a study on the particle–hole instability of Dirac fermions. We show that the formation and condensation of excitons can lead to a novel topological excitonic insulator that exhibits the parity anomaly. Furthermore, in sharp contrast to conventional semiconductors, the 2D helical Dirac liquid with spin-momentum locking (e.g., the TI surface), has an inherent tendency towards the formation of spin-triplet excitons. This remarkable property manifests itself most clearly by the interband optical absorption processes, as shown by Fig. 1.

For conventional semiconductors with absence of the spin–orbit coupling, the direct interband excitation generates a conduction electron and a valence hole with the same momentum $\mathbf{k}$ and the opposite spins (Fig. 1a), resulting in a spin-singlet exciton after they form into a pair. The situation is, however, different in TI surfaces, as shown by Fig. 1b. Due to the spin-momentum locking of TI surface, the direct interband excitation is accompanied by the spin-flip of the electron; the resultant exciton consists of an electron and a hole with the same spin polarization, and therefore a spin-triplet state. This unique optical property of TI surface, as will be shown below, can be used for generation and stabilization of a unique $p + ip$ excitonic insulator. The natural topological $p$-wave excitonic insulator has $p + ip$ symmetry inherent to their order parameters. We firstly explore the salient feature of the $p + ip$ excitonic state by studying a minimal theoretical model. In contrary to the conventional excitonic insulators[20] described by a $s$-wave Bardeen–Cooper–Schrieffe (BCS)-like theory[21], the $p + ip$ state mathematically resembles the spinless $p + ip$ SC[22] in the chiral basis. However, unlike the $p + ip$ SC, it does not break the global U(1) gauge symmetry but only violates the TRS. To characterize such exotic states, we propose a chiral Chern number $C_c$ and unveil the underlying parity anomaly by deriving an effective Chern–Simons gauge theory associated with topological defects. This is further responsible for (i) fractionalized excitations with charge $e/2$ at vortices and (ii) a spontaneous in-plane magnetization of ground state. Then, we investigate the excitonic instability of a strongly correlated TI surface model

where we show that a $p$-wave-type interaction component emerges intrinsically, which stabilizes the $p + ip$ excitonic phase in a significant parameter region of the phase diagram. Moreover, we also consider the effects of interband optical processes and we demonstrate that the electron–photon coupling can effectively drive the system into the proposed $p + ip$ excitonic insulator as a result of the spin-flip mechanism demonstrated in Fig. 1b.

## Results

**Model.** We start from a spontaneous symmetry breaking of 2D Dirac fermions with spin-momentum locking and a momentum-dependent mass term, $H = H_{\mathrm{Dirac}} + \hat{M}(\mathbf{k})$, where

$$H_{\mathrm{Dirac}} = \sum_{\mathbf{k}} \Phi_{\mathbf{k}}^{\dagger} v_{\mathrm{F}} \boldsymbol{\sigma} \cdot \mathbf{k} \, \Phi_{\mathbf{k}}. \tag{1}$$

$\mathbf{k}$ is the 2D wave vector, $\Phi_{\mathbf{k}} = [\psi_{\mathbf{k},\uparrow}, \psi_{\mathbf{k},\downarrow}]^{\mathrm{T}}$ is the fermionic spinor and $\psi_{\mathbf{k},\sigma}^{\dagger}$ ($\psi_{\mathbf{k},\sigma}$) denotes the creation (annihilation) operator of electrons with spin $\sigma$. We firstly focus on a concrete mass term $\hat{M}(\mathbf{k}) = \sum_{\mathbf{k}} \Phi_{\mathbf{k}}^{\dagger} (\Delta_{\mathrm{c}} \cos\theta \sigma^z + \Delta_{\mathrm{c}} \sin^2\theta \sigma^x - \Delta_{\mathrm{c}} \sin\theta\cos\theta\sigma^y) \Phi_{\mathbf{k}}$, while under which conditions $\hat{M}(\mathbf{k})$ can be spontaneously generated will be investigated below. $\theta$ is the polar angle of $\mathbf{k}$, and $\Delta_{\mathrm{c}}$ is the overall gap scale. It is straightforward to verify that $\hat{M}(\mathbf{k})$ breaks TRS, $\hat{T}\hat{M}(\mathbf{k})\hat{T}^{-1} \neq \hat{M}(-\mathbf{k})$, where $\hat{T} = i\sigma^y K$ being the time-reversal operator, with $K$ the complex conjugation operator. In contrast with the TI surface with a Zeeman field, where the diagonal term $m\sigma^z$ breaks TRS[23], here the diagonal term $\Delta_{\mathrm{c}} \cos\theta\sigma^z$ in $\hat{M}(\mathbf{k})$ respects the TRS. The TRS is however violated by the two off-diagonal terms in $\hat{M}(\mathbf{k})$.

One can unveil the physical meaning of $\hat{M}(\mathbf{k})$ by making a unitary transformation onto the chiral basis with $H_{\mathrm{Dirac}}$ being diagonalized: $C_{\mathbf{k}} = \hat{R}(\mathbf{k})\Phi_{\mathbf{k}}$, where $\hat{R}(\mathbf{k})$ is a 2 by 2 rotation matrix with $\hat{R}_{11} = -\hat{R}_{21} = e^{i\theta}/\sqrt{2}$, $\hat{R}_{12} = \hat{R}_{22} = 1/\sqrt{2}$, and $C_{\mathbf{k}} = [c_{\mathbf{k},+}, c_{\mathbf{k},-}]^{\mathrm{T}}$ is the spinor composed of the conduction and valence band fermions. In the chiral basis, $H$ becomes

$$H = \sum_{\mathbf{k}} \begin{pmatrix} c_{\mathbf{k},+} \\ c_{\mathbf{k},-} \end{pmatrix}^{\dagger} \begin{pmatrix} v_{\mathrm{F}}k & -\Delta_{\mathrm{c}}e^{-i\theta} \\ -\Delta_{\mathrm{c}}e^{i\theta} & -v_{\mathrm{F}}k \end{pmatrix} \begin{pmatrix} c_{\mathbf{k},+} \\ c_{\mathbf{k},-} \end{pmatrix}. \tag{2}$$

The off-diagonal $\Delta_{\mathrm{c}}e^{-i\theta}$ describes the process where an electron in the valence band is scattered up to the conduction band, leaving a particle–hole excitation on top of the many-body ground state. For $\Delta_{\mathrm{c}} \neq 0$, the particle–hole pairs condense and gap out the Dirac cone, generating an excitonic insulator. Another interesting feature of the off-diagonal term is the factor $e^{\pm i\theta} = (k_x \pm i k_y)/k$[24], similar to the spinless $p + ip$-wave SC[25], but we note that the order parameter resides in the particle–hole rather than the particle–particle basis. The model Hamiltonian can be analyzed similarly as the BCS superconductor theory. The ground state wave function $|\Psi\rangle$ is obtained after a Bogoliubov transformation as,

$$|\Psi\rangle = \prod_{\mathbf{k}} u_{\mathbf{k}} e^{\sum_{\mathbf{k}} g_{\mathbf{k}} c_{\mathbf{k},+}^{\dagger} c_{\mathbf{k},-}} |0\rangle, \tag{3}$$

with $g_{\mathbf{k}} = v_{\mathbf{k}}/u_{\mathbf{k}}$, $u_{\mathbf{k}} = \Delta_{\mathrm{c}}(k_x - ik_y)/k\xi_{\mathbf{k}}$, $v_{\mathbf{k}} = v_{\mathrm{F}}k/\xi_{\mathbf{k}}$, where $\xi_{\mathbf{k}} = \sqrt{\Delta_{\mathrm{c}}^2 + v_{\mathrm{F}}^2 k^2}$ is the energy spectrum of quasiparticles. $|\Psi\rangle$ clearly describes a coherent state of electron–hole pairs with the distribution factor $g_{\mathbf{k}}$ of the $p + ip$ symmetry.

**Topological classification and chiral Chern number.** To examine the topological classification of the $p + ip$ excitonic insulator, we write Eq. (2) into the compact form as,

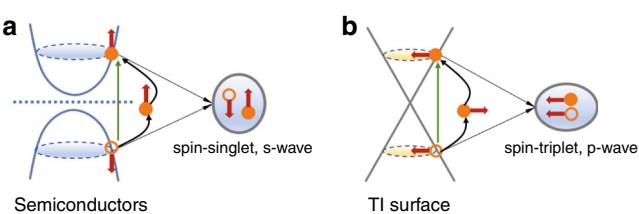

**a** Semiconductors — spin-singlet, s-wave

**b** TI surface — spin-triplet, p-wave

**Fig. 1** Schematic plot of the direct interband optical absorption process. **a** The $s$-wave excitons are formed in the conventional semiconductors. **b** The helical spin texture of topological insulator (TI) surface state favors $p$-wave rather than $s$-wave excitons

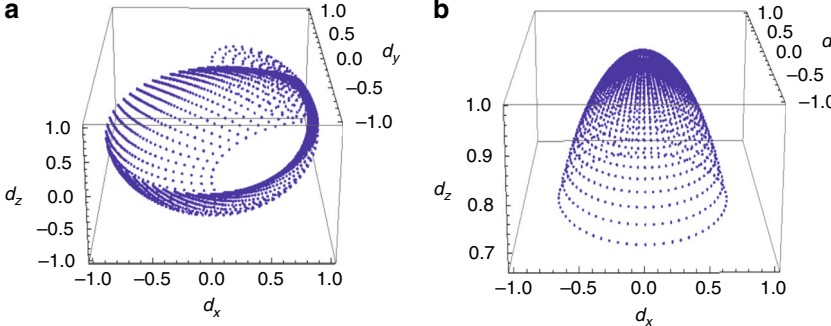

**Fig. 2** The mapping of $\hat{d}(\mathbf{k})$. **a** The mapping in the original spin basis. **b** The mapping in the chiral basis. The difference between the two situations shows that the two topological invariants are needed to completely characterize the $p + ip$ excitonic insulator, i.e., $(C, C_c) = (0, 1/2)$

$\mathcal{H} = \sum_{i=1}^{3} d(\mathbf{k})\tau^i$, with $\tau^i$ being the Pauli matrices defined in the chiral basis. For the continuum model, where $k \to \infty$ is envisioned as a single point, the three-vector $\mathbf{d}(\mathbf{k}) = (-\Delta_c \cos\theta, -\Delta_c \sin\theta, v_F k)$ defines a mapping from the $\mathbf{k}$-space ($S^2$) to the $d$-vector space. Since the $z$-component $v_F k \geq 0$, the normalized $d$-vector $\hat{d}(\mathbf{k}) = \mathbf{d}(\mathbf{k})/|\mathbf{d}(\mathbf{k})|$ resides in a hemisphere surface. For $\hat{d}(\mathbf{k})$ covering the hemisphere $n$ times, the winding number becomes $n/2$. This winding number is a topological invariant defined in the chiral basis $C_{\mathbf{k}} = [c_{\mathbf{k},+}, c_{\mathbf{k},-}]^T$, which we term as the chiral Chern number $C_c$.

Recalling that for the TI surface with a Zeeman field, a conventional Chern number $C$ is defined in the same way[16,23] but in the spin basis, $\Phi_{\mathbf{k}} = [\psi_{\mathbf{k},\uparrow}, \psi_{\mathbf{k},\downarrow}]^T$. The difference between $C$ and $C_c$ becomes clear from the perspective of Berry phase as following. In the original spin basis, following a closed loop in momentum space, the accumulation of Berry phase is $\phi = -i \oint d\mathbf{k} \cdot \langle u_{\mathbf{k}}|\nabla_{\mathbf{k}}|u_{\mathbf{k}}\rangle$, where $|u_{\mathbf{k}}\rangle$ is the eigenstate of the valence band in the spin basis. For a Bloch state defined in the chiral basis, a similar geometric phase can be defined as $\phi^c = -i \oint d\mathbf{k} \cdot \langle u_{\mathbf{k}}^c|\nabla_{\mathbf{k}}|u_{\mathbf{k}}^c\rangle$, where $|u_{\mathbf{k}}^c\rangle$ is the eigenstate of the valence band in the chiral basis. The Bloch states in the above two formalisms are related to each other by the unitary transformation $|u_{\mathbf{k}}^c\rangle = \hat{R}(\mathbf{k})|u_{\mathbf{k}}\rangle$. It then becomes obvious that the two Berry phases satisfy $\phi^c - \phi = -i \oint d\mathbf{k} \cdot \langle u_{\mathbf{k}}|\hat{R}^\dagger(\mathbf{k})\nabla_{\mathbf{k}}\hat{R}(\mathbf{k})|u_{\mathbf{k}}\rangle$, leading to $C_c - C = -(i/2\pi)\oint d\mathbf{k} \cdot \langle u_{\mathbf{k}}|\hat{R}^\dagger(\mathbf{k})\nabla_{\mathbf{k}}\hat{R}(\mathbf{k})|u_{\mathbf{k}}\rangle = 1/2$, after inserting the $\hat{R}(\mathbf{k})$ matrix we used to diagonalize $H_{\text{Dirac}}$.

To explicitly demonstrate the topological invariants, we plot the normalized vector $\hat{d}(\mathbf{k})$ with traversing the whole $\mathbf{k}$-space. Figure 2a, b shows the mapping of the $\hat{d}(\mathbf{k})$ vector in the spin and chiral basis, respectively. It is found that $\hat{d}(\mathbf{k})$ in latter basis completely covers the hemisphere while $\hat{d}(\mathbf{k})$ in the former does not, leading to $C_c = 1/2$ and $C = 0$. Therefore, two topological invariants are needed to completely characterize the $p + ip$ excitonic state, namely, $(C, C_c) = (0, 1/2)$. The zero Chern number $C = 0$ means that the $p + ip$ excitonic insulator is topologically distinct from the TI surface with a Zeeman field that has $C = 1/2$[16]. This is clear from the Landau quantization of the two states. For the TI surface with a Zeeman field, the Landau level (LL) for $|n| \geq 1$ reads as $E_n = \pm\sqrt{2ev_F^2 B|n| + m^2}$, where $m$ is the Zeeman gap. In addition, there is an unconventional zeroth LL $E_0 = \text{sgn}(m)m$ which is located exactly at the gap edge[26–28]. This zeroth LL is a manifestation of the nontrivial Zeeman mass as a result of $C = 1/2$[27,28]. For the $p + ip$ excitonic insulator phase, however, one cannot find the unconventional zeroth LL located at the gap edge, consistent with $C = 0$.

**Chern–Simons theory and fractionalization.** The nonzero chiral Chern number $C_c = 1/2$ indicates that the $p + ip$ excitonic insulator must be topologically distinct from normal insulators.

Recalling the $p + ip$ SCs where Majorana fermions are encoded in vortices[22], we shift our attention to topological defects in the $p + ip$ excitonic state. To this end, we can solve the Bogoliubov de Gennes (BdG)-type equations looking for zero-energy solutions. In real space, the equations can be derived as (see Methods for details):

$$e^{i\alpha}\left(-i\partial_r + r^{-1}\partial_\alpha\right)u(\mathbf{r}) - \frac{1}{v_F}\Delta_c(\mathbf{r})v(\mathbf{r}) = 0, \qquad (4)$$

$$e^{-i\alpha}\left(-i\partial_r - r^{-1}\partial_\alpha\right)v(\mathbf{r}) + \frac{1}{v_F}\Delta_c^*(\mathbf{r})u(\mathbf{r}) = 0, \qquad (5)$$

where $\alpha$ is the polar angle in real space and $u$, $v$ are the two components of the spinor wave function. Similar equations have been studied in refs. [22,29,30]. However, note that the physics in the present model is different from previous works. Read and Green[22] derive the equations for SCs, where charge conservation is broken and Majorana fermions are the emergent excitations. For our system, the U(1) gauge symmetry is preserved, therefore no Majorana modes are allowed. Hou et al.[29] investigate graphene with a chirality mixing texture which induces the mass term, and Seradjeh et al.[30] discuss the exciton condensation between two opposite TI surfaces. In contrary, here we focus on the excitonic order in a Dirac liquid state and the mass term is generated by the condensation of $p + ip$ excitons. To solve the equations in the presence of a topological defect[31], we assume $\Delta_c(\mathbf{r}) = \Delta_c(r)e^{i\alpha}$. A unique normalized zero-energy bound state[32,33] can be obtained as, $u = Af(r)e^{i(\alpha + \pi/2)/2}$ and $v = u^*$, where $A$ is the normalization constant and the function $f(r) = \exp[-|\Delta_c|r/|v_F|]$ decays exponentially.

We note that the single particle Hamiltonian $h$ in Eq. (2) satisfies $(i\tau^2)h(i\tau^2) = h$. This allows one to define a time-reversal-type operator $\Theta = i\tau^2 K$, with which it is clear that for any eigenstate $\psi_E$ with energy $E$, $\Theta\psi_E$ is also a eigenstate with energy $-E$. Besides, the total number of fermion states are conserved, namely, $\int_{-\infty}^{\infty} dE\delta\rho(E) = 0$, where $\delta\rho(E)$ is the change of density of states. Applying this formula to the case where a topological defect is in presence, and then taking into account the fact that $\rho(E)$ is a symmetric function with respect to $E = 0$, one comes to the conclusion that both the conduction and valence band lose half a state. This state is compensated by the zero mode that is bound to the topological defect. Consequently, at half filling, the electron charge associated to the defect is either $-e/2$ or $e/2$, depending on whether the zero mode is occupied or empty.

We have shown that the $p + ip$ excitonic insulator is characterized by $C_c = 1/2$ and displays fractional charge $e/2$ at vortices. However, it is still unclear how the two observations are related to each other. This can be answered by the parity anomaly[34] of the $p + ip$ excitonic insulator and the effective gauge

theory associated with the topological defects. First, we couple an external U(1) gauge field $A_\mu$ (with $\mu = 0, 1, 2$) to the $p + ip$ excitonic state[35], and we introduce the Dirac matrices defined as $\gamma^0 = \tau^3$, $\gamma^{1,2} = i\tau^{1,2}$. In order to extract the effective theory for the vortices, we include the vortex degrees of freedom $\chi$ in the order parameter. Second, we perform a singular gauge transformation, $c_\pm \to e^{\pm i\chi/2} c_\pm$[36], which results in the following Lagrangian (see Methods for details),

$$\mathcal{L} = \overline{\mathcal{C}}_{\mathbf{k}} \left( i\tilde{\gamma}^\mu \partial_\mu + \tilde{\gamma}^\mu a_\mu + m \right) \mathcal{C}_{\mathbf{k}}, \tag{6}$$

where $\mathcal{C}_{\mathbf{k}} \left( \overline{\mathcal{C}}_{\mathbf{k}} \right)$ is the Grassmann field. $\tilde{\gamma}^\mu$ are defined as $\tilde{\gamma}^0 = \gamma^0$, $\tilde{\gamma}^1 = \gamma^2$, $\tilde{\gamma}^2 = -\gamma^1$. In Eq. (6), the vortex degrees of freedom have been absorbed into an effective gauge field $a_\mu$ after the singular transformation. For a static defect with $\partial_0 \chi = 0$, $a_\mu = A_\mu + \frac{1}{2} \partial_\mu \chi$. Finally, after one integrates out the Grassmann field, a Chern–Simons term with respect to the effective gauge field $a_\mu$ can be obtained in the second order expansion:

$$\mathcal{L}_{\text{CS}} = \frac{I}{2\pi} \epsilon^{\mu\nu\rho} a_\mu \partial_\nu a_\rho, \tag{7}$$

where the coefficient $I$ is found to be $1/2$ for the proposed $p + ip$ excitonic insulator.

We note that for the TI surface with a Zeeman field[23], a similar Chern–Simons Lagrangian arises as a result of the parity anomaly, where the coefficient in front of the Chern–Simons term is exactly the Chern number $C$. Here, the coefficient $I = 1/2$ is identified to be the chiral Chern number $C_c$ defined above. The similarity between the two gauge field theories is due to the underlying parity anomaly; however, for the $p + ip$ excitonic insulator, it is the effective gauge field $a_\mu$ rather than the external field $A_\mu$ that obeys the Chern–Simons form. Further inserting $a_\mu = A_\mu + \frac{1}{2} \partial_\mu \chi$, the full Lagrangian $\mathcal{L}_{\text{CS}} = \frac{I}{2\pi} \epsilon^{\mu\nu\rho} A_\mu \partial_\nu A_\rho - A_\mu j^\mu + \mathcal{L}_v$ consists of a Lagrangian describing the vortex $\mathcal{L}_v$ and a current operator associated with the vortex,

$$j^\mu = \frac{I}{2\pi} \epsilon^{\mu\nu\rho} \partial_\nu \partial_\rho \chi. \tag{8}$$

For a static vortex (with vortex number $n = 1$) that centered at $\mathbf{r}_0$, we have $\epsilon^{ij} \partial_i \partial_j \chi = 2\pi \delta(\mathbf{r} - \mathbf{r}_0)$, inserting which into Eq. (8), the electron charge of the defect is then obtained as $Q = \int d^2 x j^0 = I \equiv C_c = 1/2$. Therefore, we have proved that the chiral Chern number $C_c$ equals to the fractional charge (in unit of e) of a topological defect, justifying $C_c$ to be the essential topological invariant characterizing the $p + ip$ excitonic insulator.

**Spontaneous in-plane magnetization.** The order parameter $-\Delta_c e^{-i\theta}$ in Eq. (2) breaks TRS. This suggests that the ground state carries a magnetization associated with it. Before proceeding, we note that the order parameter $-\Delta_c e^{-i\theta}$ still enjoys a redundant global phase. In the following, we take this into account and write the off-diagonal term in Eq. (2) as $-\Delta_c e^{-i\theta'}$, where $\theta' = \theta - \beta$ with $\beta$ being a constant angle independent of $\mathbf{k}$. The global phase $\beta$ is analogous to U(1) phase of a superconductor.

To clearly show the feature of magnetization, we calculate the spin expectation values $\langle \sigma^x \rangle$, $\langle \sigma^y \rangle$, $\langle \sigma^z \rangle$ from the single particle eigenstates and study the spin texture of the $p + ip$ excitonic ground state $|\Psi\rangle$ (Eq. (3)). We obtain

$$\langle \sigma^x \rangle = -\frac{(\Delta_c \cos\theta' + \xi_\mathbf{k})(v_F k \cos\theta' + \Delta_c \sin^2\theta')}{v_F^2 k^2 + \Delta_c^2 + \Delta_c \cos\theta' \xi_\mathbf{k}},$$

$$\langle \sigma^y \rangle = -\frac{(\Delta_c \cos\theta' + \xi_\mathbf{k})(v_F k \sin\theta' - \Delta_c \sin\theta' \cos\theta')}{v_F^2 k^2 + \Delta_c^2 + \Delta_c \cos\theta' \xi_\mathbf{k}},$$

$$\langle \sigma^z \rangle = -\frac{\Delta_c^2 \sin^2\theta' - \Delta_c^2 \cos^2\theta' - \Delta_c^2 - 2\Delta_c \cos\theta' \xi_\mathbf{k}}{v_F^2 k^2 + \Delta_c^2 + \Delta_c \cos\theta' \xi_\mathbf{k}}.$$

As has been discussed before, $\langle \sigma^z \rangle$ does not break TRS. The integral of $\langle \sigma^z \rangle$ over the whole $\mathbf{k}$-space vanishes. The in-plane spin texture ($\langle \sigma^x \rangle$, $\langle \sigma^y \rangle$) is however nonzero, and we plot it in Fig. 3 for different values of $\beta$, i.e., $\beta = 0$, $\beta = \pi/2$, $\beta = \pi$, $\beta = 3\pi/2$. Interestingly, it is found that the in-plane spin configuration is tilted from the standard helical spin texture of the original TI surface, and the spins at two TRS-related momentum points, $\mathbf{k}$ and $-\mathbf{k}$, are no longer always opposite to each other, violating the TRS. Though being tilted, the chiral nature of electron's spin is preserved as we can see that the direction of the vector ($\langle \sigma^x \rangle$, $\langle \sigma^y \rangle$) still undergoes a change of $2\pi$ when traversing a closed path around $\mathbf{k} = 0$ point. Furthermore, we calculate the total in-plane magnetization $\mathbf{M}$ by integrating ($\langle \sigma^x \rangle$, $\langle \sigma^y \rangle$) over the momentum space. It is found that $M$ always points towards the opposite direction of the global phase $\beta$, as shown by the thick red arrows in Fig. 3. We also note that the emergent magnetization of ground state can lead to anisotropic in-plane transport of excitons which serves as an evident experimental signature of the $p + ip$ excitonic insulator. Furthermore, since $\beta$ determines the direction of magnetization of ground state, one expects to control $\beta$ by applying a weak in-plane magnetic field during the process of TRS symmetry breaking.

**The excitonic phases on interacting TI surfaces.** After careful characterization of the topological properties of the $p + ip$ excitonic insulator, we now study its possible realization on interacting TI surface states. Although various instabilities of interacting topological insulators and topological superconductors have been extensively studied[37–48], we focus on the excitonic instability of the surface states and propose to realize the minimal model of the topological excitonic insulators via spontaneous symmetry breaking that enjoys a fractional chiral Chern number $C_c = 1/2$, which is different from earlier proposals based on semiconductor heterostructure with an integer Thouless–Kohmoto–Nightingale–Nijs (TKNN) number[49].

The effective model Eq. (1) describing the surface state is usually modified by a higher order wrapping effect, reducing the rotational symmetry of the Dirac cone down to a discrete $C_3$ subgroup. Our further studies on the wrapping effect shows that the exciton gap remains very robust and therefore one can neglect the wrapping effect for the following analysis. We firstly consider the particle–hole symmetric case, while the generalization to finite chemical potential $\mu$ is very straightforward and will be discussed below.

First we start with demonstrating that a $p$-wave-type interaction that spontaneously generates the $p + ip$-wave exciton insulator naturally arises from the Hubbard model, which reads in $\mathbf{k}$-space as:

$$H_I = \frac{V}{2} \sum_{\mathbf{k},\mathbf{k}',\mathbf{q}} \sum_\sigma \psi^\dagger_{\mathbf{k}+\mathbf{q},\sigma} \psi^\dagger_{\mathbf{k}'-\mathbf{q},\overline{\sigma}} \psi_{\mathbf{k}',\overline{\sigma}} \psi_{\mathbf{k},\sigma}. \tag{9}$$

To facilitate the study of excitonic instability, we transform $H_I$ onto the chiral basis. Then, considering only interband forward scattering channel, the low-energy effective vectorial interaction component can be derived as (see Methods for details),

$$H = \sum_{\mathbf{k},n} n v_F k c^\dagger_{\mathbf{k},n} c_{\mathbf{k},n}$$
$$- \frac{V}{4} \sum_{\mathbf{k},\mathbf{k}',n} \left[ \hat{k} c^\dagger_{\mathbf{k},n} c_{\mathbf{k},-n} \right] \cdot \left[ \hat{k}' c^\dagger_{\mathbf{k}',-n} c_{\mathbf{k}',n} \right], \tag{10}$$

where we assume a momentum cutoff $\Lambda$ implicit in the sum of $\mathbf{k}$. $n$ is the band index. The interaction term in the second line, $H^{(1)}_{\text{eff}} = -\frac{V}{4} \sum_{\mathbf{k},\mathbf{k}',n} \left[ \hat{k} c^\dagger_{\mathbf{k},n} c_{\mathbf{k},-n} \right] \cdot \left[ \hat{k}' c^\dagger_{\mathbf{k}',-n} c_{\mathbf{k}',n} \right]$, is of $p$-wave type,

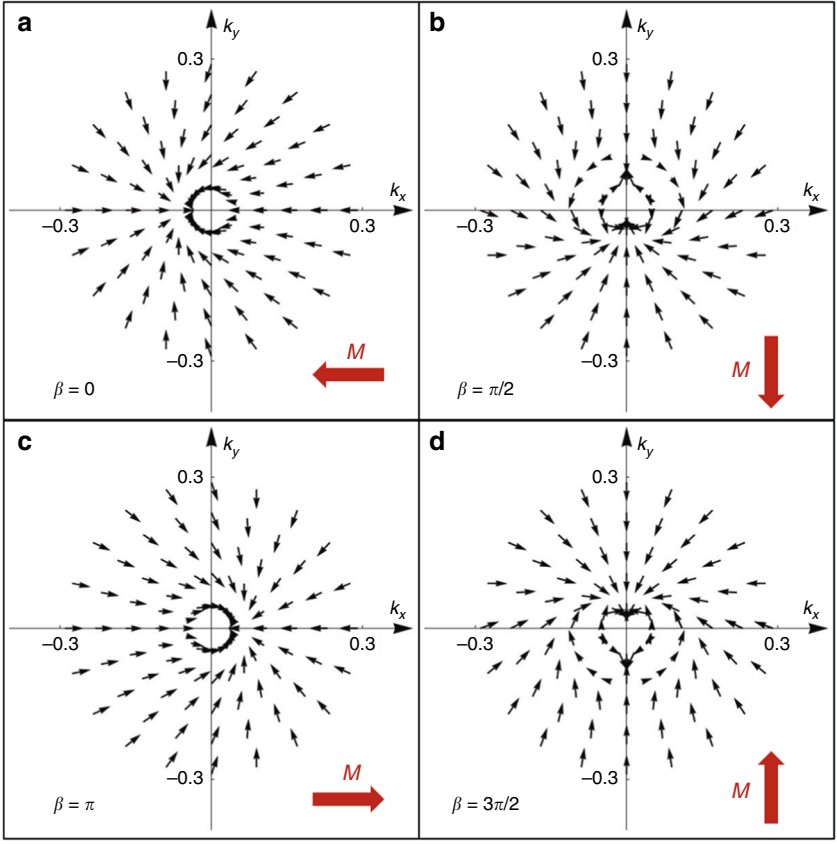

**Fig. 3** The in-plane spin texture of the $p + ip$ excitonic insulator. **a–d** The results for different global phase of the exciton order parameter, $\beta = 0$, $\beta = \pi/2$, $\beta = \pi$, $\beta = 3\pi/2$, respectively. The total net magnetization $M$ is shown by the arrow for each case

and it arises from the spin-momentum locking of the surface state. After introducing a complex bosonic vector field $\boldsymbol{\Delta}$ and performing the Hubbard–Stratonovich transformation to decouple $H_{\text{eff}}^{(1)}$, we arrive at a mean-field theory with the order parameter given by $\boldsymbol{\Delta} = \frac{1}{2} V \sum_{\mathbf{k}} \left\langle \hat{k} c_{\mathbf{k},-}^{\dagger} c_{\mathbf{k},+} \right\rangle$. When $\boldsymbol{\Delta} \neq 0$, the condensation of excitons gaps out the Dirac point, resulting in the excitonic insulator phase. Similar to $^3$He superfluid theory[25], symmetry analysis implies two possible configurations of the order parameter that are the extrema of the free energy: (1) (chiral) $p + ip$ excitonic insulator with $\boldsymbol{\Delta} = \frac{1}{\sqrt{2}} \Delta_{\text{c}} \left( \widehat{\mathbf{e}}_x - i \widehat{\mathbf{e}}_y \right)$ and (2) the polar excitonic insulator with $\boldsymbol{\Delta} = \Delta_{\text{p}} \widehat{\mathbf{e}}$, where $\widehat{\mathbf{e}}$ is a unit vector in the xy plane. The polar excitonic phase is topologically trivial with zero chiral Chern number.

By self-consistently solving the mean-field gap equation, we obtain the condensation energy and the order parameter $(\Delta_{\text{c}}, \Delta_{\text{p}})$, which are plot in Fig. 4a, b, respectively. Figure 4b shows the emergence of a threshold $V_c$, beyond which the system develops the excitonic instabilities with $|\Delta_{\text{c/p}}| \neq 0$. Figure 4a indicates that even though both the $p + ip$ excitonic insulator and polar excitonic insulator are stable compared to the gapless Dirac cone for $V > V_c$, the $p + ip$ excitonic insulator is more energetically favored due to the lower condensation energy. Moreover, we also performed addition mean-field calculations by treating the $p + ip$ and polar excitonic instability on equal footing. It is found that the $p + ip$ channel always suppresses the polar order down to zero. Therefore, even though the ground state energies of the two states are close, the mass term of the polar exciton insulator is negligible in the mean-field theory. Hence, the resultant mean-field Hamiltonian exactly matches with Eq. (2), and the proposed mass term $\hat{M}(\mathbf{k})$ is generated by spontaneous TRS breaking as a result of the $p$-wave interaction $H_{\text{eff}}^{(1)}$. Moreover, we note that for particle–hole asymmetric case with a finite chemical potential $\mu$, the critical $V_c$ increases with increasing $\mu$. Although the exciton insulator is always stable for strong enough interaction $V > V_c$, it is more experimentally favorable to tune $\mu$ close to the Dirac point[50,51].

In the above calculations, we temporarily focused on the $p$-wave interaction $H_{\text{eff}}^{(1)}$ without considering the $s$-wave component. Nevertheless, the results clearly show that the TI surface state indeed has a strong tendency towards the $p + ip$ excitonic instability via spontaneous TRS breaking. We now take into account the interaction in the $s$-wave channel, which can also be derived from the local Hubbard interaction as (see Methods for details), $H_{\text{eff}}^{(2)} = -\frac{V}{4} \sum_{\mathbf{k},\mathbf{k}',n} c_{\mathbf{k},n}^{\dagger} c_{\mathbf{k},-n} c_{\mathbf{k}',-n}^{\dagger} c_{\mathbf{k}',n}$. In order to be more general, we assume in the following two independent couplings $V_{\text{p}}$ and $V_{\text{s}}$ for the $p$-wave and $s$-wave interaction respectively, and treat both terms on equal footing. An $s$-wave bosonic scalar field $\Delta_{\text{s}} = V_{\text{s}} \sum_{\mathbf{k}} \left\langle c_{\mathbf{k}-}^{\dagger} c_{\mathbf{k}+} \right\rangle$ is introduced to decouple $H_{\text{eff}}^{(2)}$ in addition to the $p + ip$ excitonic order parameter $\Delta_{\text{c}}$. The coupled self-consistent equations with respect to $\Delta_{\text{s}}$ and $\Delta_{\text{c}}$ are obtained in the standard approach which leads to the calculated phase diagram shown by Fig. 4c. It is found that both the $s$-wave and $p + ip$-wave excitonic insulator can be spontaneously generated from the Dirac liquid phase with strong enough interactions. The former and latter are stabilized for large $V_{\text{s}}$ and $V_{\text{p}}$ respectively. We note that although the $p + ip$ excitonic phase, which is absent in conventional semiconductors, becomes stable in a large parameter region of the phase diagram, it does not occur as the ground state for the discussed local Hubbard model, which lies in the parameter line $V_{\text{s}} = V_{\text{p}}$ that only enters into the $s$-wave excitonic condensation.

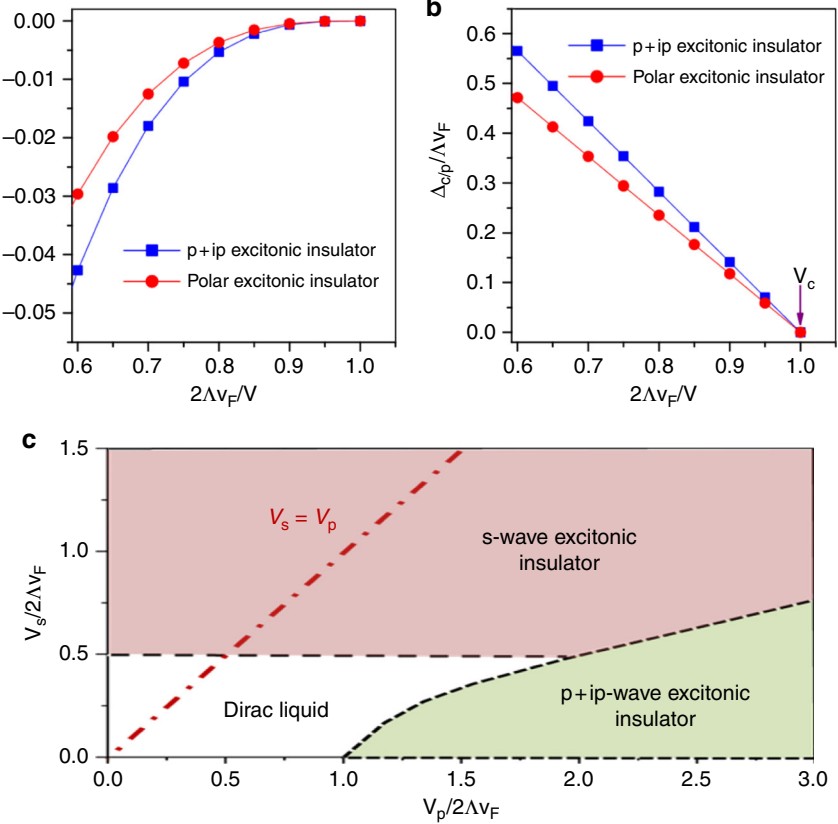

**Fig. 4** The results from mean-field calculations. **a** Condensation energy versus interaction $V$ for $p + ip$ excitonic insulator and polar excitonic insulator. **b** Calculated mean-field excitonic order parameter for $p + ip$ excitonic insulator ($\Delta_c$) and polar excitonic insulator ($\Delta_p$) versus interaction $V$. $\left|\Delta_{c/p}\right|^2/v_F^2\Lambda^2 \ll 1$ is assumed. **c** The phase diagram based on unbiased mean-field with considering both the $s$- and $p + ip$-wave interaction channel. The dash-dotted red line denotes the Hubbard interaction case with $V_s = V_p$

**The stabilization of the $p + ip$ excitonic insulator**. Now we study the stabilization of the $p + ip$ excitonic insulator on TI surfaces by going beyond the natural Dirac semimetals and taking into account external perturbations. Recalling the interband optical absorption process shown by Fig. 1b, one expects that additional photons should prefer the $p$-wave pairing due to the spin-flip mechanism that gives rise to spin-triplet excitons. Since excitons are usually generated by absorption of photons in a material, we consider a linearly polarized perpendicular incident laser with frequency $\bar\omega$ and light intensity $I$. The corresponding microscopic theory can be constructed by quantization of the vector potential $\mathbf{A}(\mathbf{r})$, i.e.,

$$\mathbf{A}(\mathbf{r}) = \sum_{\mathbf{q}\lambda} \sqrt{\frac{2\pi}{\omega_{\mathbf{q}}}} \mathbf{e}_\lambda \left(a_{\mathbf{q},\lambda} + a^\dagger_{-\mathbf{q},\lambda}\right) e^{-i\mathbf{q}\cdot\mathbf{r}}, \tag{11}$$

where $a_{\mathbf{q},\lambda}$ $\left(a^\dagger_{\mathbf{q},\lambda}\right)$ is the annihilation (creation) operator of photons, $\mathbf{e}_\lambda$ ($\lambda = 1, 2$) are the polarization vectors, $\omega_{\mathbf{q}} = c|\mathbf{q}|$ is the light frequency, with $c$ being the light velocity and $\mathbf{q}$ the wave vector of photons.

The coupling of the external field to the surface electrons is described within the formalism of the minimal coupling, which leads to the electron–photon interaction,

$$H_{\text{el-ph}} = -ev_F \sum_{\mathbf{kq}\lambda\sigma\sigma'} \sqrt{\frac{2\pi}{\omega_{\mathbf{q}}}}(\boldsymbol{\sigma}_{\sigma\sigma'}\cdot\mathbf{e}_\lambda)\left(\psi^\dagger_{\mathbf{k},\sigma}a_{\mathbf{q},\lambda}\psi_{\mathbf{k}-\mathbf{q},\sigma'}\right.$$
$$\left. +\psi^\dagger_{\mathbf{k},\sigma}a^\dagger_{-\mathbf{q},\lambda}\psi_{\mathbf{k}-\mathbf{q},\sigma'}\right), \tag{12}$$

where the term $(\boldsymbol{\sigma}_{\sigma\sigma'}\cdot\mathbf{e}_\lambda)$ arises due to the helical spin texture and

the spin-momentum locking of TI surface states. For linearly polarized perpendicular incident photons, $\mathbf{e}_1 = \widehat{\mathbf{e}}_x$ and $\mathbf{e}_2 = \widehat{\mathbf{e}}_y$. Thus, the spin-flip matrices $\sigma^x$ and $\sigma^y$ take place in $H_{\text{el-ph}}$. This verifies the spin-flip mechanism illustrated by Fig. 1b in a microscopic perspective. Moreover, since $c \gg v_F$, the conservation of energy and momentum requires $|\mathbf{k}| \gg |\mathbf{q}|$ in $H_{\text{el-ph}}$, and therefore ensures the direct interband excitation with $\mathbf{k} - \mathbf{q} \simeq \mathbf{k}$. Furthermore, the photon fields are Gaussian with the Hamiltonian $H_{\text{ph}} = \sum_{\mathbf{q}\lambda} \omega_{\mathbf{q}}\left(a^\dagger_{\mathbf{q},\lambda}a_{\mathbf{q},\lambda} + 1/2\right)$, one can therefore exactly integrate them out. This procedure leads to the effective action $S_{\text{int}}$ describing the renormalization from photons:

$$S_{\text{int}} = -\int dt \sum_{\mathbf{kk}'\mathbf{q}} \sum_{\alpha\beta\rho\gamma\lambda} \frac{2\pi e^2 v_F^2}{\omega_{\mathbf{q}}}\left(\boldsymbol{\sigma}_{\alpha\beta}\cdot\mathbf{e}_\lambda\right)\left(\boldsymbol{\sigma}_{\rho\gamma}\cdot\mathbf{e}_\lambda\right)$$
$$\times \frac{1}{i\partial_t - \omega_{\mathbf{q}}}\psi^\dagger_{\mathbf{k},\alpha}\psi_{\mathbf{k},\beta}\psi^\dagger_{\mathbf{k}',\rho}\psi_{\mathbf{k}',\gamma}. \tag{13}$$

Where $\sum_\lambda \left(\boldsymbol{\sigma}_{\alpha\beta}\cdot\mathbf{e}_\lambda\right)\left(\boldsymbol{\sigma}_{\rho\gamma}\cdot\mathbf{e}_\lambda\right) = \sigma^x_{\alpha\beta}\sigma^x_{\rho\gamma} + \sigma^y_{\alpha\beta}\sigma^y_{\rho\gamma}$, inserting which it is found that $S_{\text{int}}$ consists of an equal-time renormalization term that produces the effective interaction between electrons,

$$H_{\text{int}} = -V_{\text{eff}} \sum_{\mathbf{kk}'\sigma} \psi^\dagger_{\mathbf{k},\sigma}\psi_{\mathbf{k},\bar\sigma}\psi^\dagger_{\mathbf{k}',\bar\sigma}\psi_{\mathbf{k}',\sigma}, \tag{14}$$

where $V_{\text{eff}} = 4\pi e^2 v_F^2 n_{\text{ph}}/\bar\omega^2$ and $n_{\text{ph}}$ is the number density of photons that interact with the electrons, which is proportional to the light intensity $I$. We have utilized the fact that all the wave vectors $\mathbf{q}$ satisfy $\mathbf{q} = q\widehat{\mathbf{e}}_z = \bar\omega\widehat{\mathbf{e}}_z/c$ for photons with the frequency $\bar\omega$. The first

two operators in $H_{\text{int}}$, $\psi_{\mathbf{k},\sigma}^{\dagger}\psi_{\mathbf{k},\overline{\sigma}}$, are generated by the optical interband absorption process with spin-flip depicted by Fig. 1b. Similarly, the last two operators, $\psi_{\mathbf{k}',\overline{\sigma}}^{\dagger}\psi_{\mathbf{k}',\sigma}$, are originated from the interband relaxation process with spin-flip. After a unitary transformation to the chiral basis, Eq. (14) is exactly cast into:

$$H_{\text{int}} = -\frac{V_{\text{eff}}}{2}\sum_{\mathbf{k},\mathbf{k}',n}\left[\hat{k}c_{\mathbf{k},n}^{\dagger}c_{\mathbf{k},-n}\right]\cdot\left[\hat{k}'c_{\mathbf{k}',-n}^{\dagger}c_{\mathbf{k}',n}\right]. \quad (15)$$

Interestingly, $H_{\text{int}}$ enjoys the $p$-wave factor $\hat{k}\cdot\hat{k}'$, exactly the same as the $p$-wave interaction component reduced from the Hubbard model, $H_{\text{eff}}^{(1)}$. The interband optical processes automatically select the $p$-wave channel due to the spin-flip scatterings (Fig. 1b). More importantly, $H_{\text{int}}$ has a strength $V_{\text{eff}}$ tunable by external field parameters, $\bar{\omega}$ and $I$. According to the mean-field study above and the calculated phase diagram in Fig. 4, it is known that the tunable external field can renormalize the Dirac cone state and drive it into the proposed $p + ip$-wave excitonic insulator. The resultant state displays the predicted fractional charges associated with vortex excitations, the spontaneous in-plane magnetization and thus the anisotropic transport of excitons on the TI surface.

## Discussion

Excitonic insulators require strong interactions since the topological surface states become unstable only when the interaction is as large as the ultraviolet cutoff (band gap) as shown in Fig. 4b. Therefore, we argue that topological Kondo insulators (TKIs) are good candidates for excitonic instability on TI surface. For instance, in the case of $SmB_6$, which is the archetypal TKI, the renormalized Coulomb interaction ($U \sim 1$ eV) is much larger than the direct band gap ($\Lambda v_{\text{F}} \sim 20$ meV). For these parameters, our mean-field theory predicts an excitonic gap of order $\Delta_{s/c} \simeq 2$–$10$ meV. Indeed, magnetism and hysteresis in magnetotransport have been observed in $SmB_6$[52]. Whether these experiments indeed observe an excitonic insulator require future work. We would like to note that our theory is not related with the recent works on excitonic contributions to quantum oscillations[53] or the intervalley excitonic instabilities[54] in $SmB_6$.

In addition to the short-range interaction, we also considered a long-range Coulomb interaction on the TI surface and make generalization to finite temperature[55]. It is found that the critical interaction $V_c$ for BCS exciton condensation is significantly reduced for long-range interaction. At finite temperature, taking into account the thermal fluctuation, the strict long-range order is absent, the Kosterliz–Thouless transition temperature $T_{\text{KT}}$ can be calculated by evaluating the phase stiffness of the exciton order based on the mean-field calculations[56]. Experimentally, the driven TI surface offers a rare platform for realization of new optical devices and facilitates the study of possible collective instabilities of Dirac electrons. Experiments measuring bulk metallic TI materials have obtained the lifetime of the excited states to be around a few picoseconds[57–59], which is significantly larger than than that in graphene. Moreover, a remarkably long lifetime exceeding 4 μs has been observed by experiment for the surface states in bulk insulating $Bi_2Te_2Se$[60]. These indicate promising observation of a transient or quasi-equilibrium excitonic insulator[59,61]. Besides, the surface has a reduced dielectric constant which can be tuned even smaller by doping or gating[62], and this possibly makes the long-range interaction more effective.

To summarize, we proposed a new type of topological $p + ip$ excitonic insulator that achieves the parity anomaly and is characterized by the nontrivial chiral Chern number. We

provided a concrete example and discussed its possible realization in strongly correlated TI surface. Interestingly enough, similar to the TIs due to band inversion, the predicted topological behaviors, i.e., the charge fractionalization and the emergent magnetization of ground state, are robust and immune to continuous deformation of the Hamiltonian, as long as the exciton gap remains unclosed. This suggests that realizations of topological $p$-wave excitonic insulators are likely in many other physical systems. For example, the spectrum under study can be continuously deformed from the Dirac cone to a quadratic band crossing point (QBCP)[63]. This indicates that the instability is also possible and may be more feasible in interacting 2D electron gas with spin-momentum locking and QBCP. In this sense, the present results are general and may provide new physical settings for the realization of quantum anomalies in condensed matter system, and may stimulate different directions to search for more exotic topological state of matters. We also note that a laser-induced non-equilibrium version of our proposal has been experimentally observed recently in $Bi_2Se_3$[55], where a millimeter-long transport distance of excitons up to 40 K is reported, supporting an excitonic condensation state formed on the TI surface. Our study on the $p + ip$ excitonic insulator stabilized by the optical pumping further predicts an anisotropic transport of excitons, which we believe would be of great interest for future experimental investigations.

## Methods

**The zero-energy BdG-type equations.** From Eq. (2) in the main text, the zero-energy eigen-equations can be written down as:

$$v_{\text{F}}ku_{\mathbf{k}} - \Delta(\mathbf{r})e^{-i\theta}v_{\mathbf{k}} = 0,$$
$$-e^{i\theta}\Delta^*(\mathbf{r})u_{\mathbf{k}} - v_{\text{F}}kv_{\mathbf{k}} = 0.$$

Here, we assume a real-space dependence of the order parameter $\Delta(\mathbf{r})$. With a vortex, we have $\Delta(\mathbf{r}) = \Delta_c(r)e^{i\alpha}$, where $\alpha$ is the polar angle of $\mathbf{r}$ in real space. To simplify the eigen-equations, one can multiply a factor $e^{i\theta}$ from the right in the first equation, and a factor $e^{-i\theta}$ from the left in the second equation. Then, a Fourier transformation yields,

$$\left(-i\partial_x + \partial_y\right)u_{\mathbf{r}} - \frac{1}{v_{\text{F}}}\Delta(\mathbf{r})v_{\mathbf{r}} = 0,$$
$$\frac{1}{v_{\text{F}}}\Delta^*(\mathbf{r})u_{\mathbf{r}} + \left(-i\partial_x - \partial_y\right)v_{\mathbf{r}} = 0.$$

After transformation to the polar coordinate, we arrive at the zero-energy BdG-like equations in real space.

$$e^{i\alpha}(-i\partial_r + r^{-1}\partial_\alpha)u(\mathbf{r}) - \frac{1}{v_{\text{F}}}\Delta_c(r)v(\mathbf{r}) = 0,$$
$$e^{-i\alpha}(-i\partial_r - r^{-1}\partial_\alpha)v(\mathbf{r}) + \frac{1}{v_{\text{F}}}\Delta_c^*(r)u(\mathbf{r}) = 0.$$

**Effective gauge field theory.** In the second quantized form, the effective Hamiltonian of the $p + ip$ excitonic insulator reads as

$$H = \sum_{\mathbf{k}} C_{\mathbf{k}}^{\dagger}\left[v_{\text{F}}k\tau^z - \Delta_c e^{-i\theta}\tau^+ - \Delta_c^* e^{i\theta}\tau^-\right]C_{\mathbf{k}}.$$

Formally, one can make transformation $C_{\mathbf{k}} \to \mathcal{C}_{\mathbf{k}} = C_{\mathbf{k}}/\sqrt{k}$, while keeping the total Hamiltonian unchanged. The action in the functional representation is then cast into:

$$S = \int d\tau d\mathbf{k}\overline{\mathcal{C}}_{\mathbf{k}}\left[i\partial_0 + \tau^z v_{\text{F}}k^2 + \Delta_c k'\gamma^- + \Delta_c^* k'^* \tau^+\right]\mathcal{C}_{\mathbf{k}},$$

where we have defined $k' = k_x - ik_y$ and introduced the Dirac matrices in 2D, $\gamma^0 = \tau^3$, $\gamma^1 = i\tau^1$, $\gamma^2 = i\tau^2$, and $\gamma^{\pm} = (\gamma^2 \pm i\gamma^1)/2$. The above action formally describes a 2D Dirac fermion with a mass $v_{\text{F}}k^2$. Since the sign of mass determines the topology of 2D Dirac systems, and $v_{\text{F}}k^2 \geq 0$ for all $k$, one can continuously deform the mass into a constant $m$ while keeping the topological property unchanged. With a U(1) gauge transformation of the Grassmann field, the gauge invariance requires the minimal coupling $\tilde{k}' = k_x - A_x - i(k_y - A_y)$. With further including the vortex degrees of freedom in the order parameter, the Lagrangian can be written as,

$$\mathcal{L} = \overline{\mathcal{C}}_{\mathbf{k}}\left(\gamma^0\widetilde{i\partial_0} + \gamma^+ e^{i\chi}\tilde{k}' + \gamma^- \tilde{k}' e^{-i\chi} + m\right)\mathcal{C}_{\mathbf{k}},$$

where $\chi$ is the vortex phase which goes from 0 to $2\pi$ when one completes a closed loop in real space. Without losing generality, we set $|\Delta_c| = 1$. The behavior of vortex degrees of freedom can be extracted from the above Lagrangian after integrating out the Grassmann fields. To the second order expansion, the Chern–Simons action in Eq. (7) is obtained.

**Mean-field analysis of the Hubbard interaction**. Let us denote the four $\mathbf{k}$ vectors in the Hubbard interaction Eq. (9) as $\mathbf{k}_1 = \mathbf{k}$, $\mathbf{k}_2 = \mathbf{k}'$, $\mathbf{k}_3 = \mathbf{k}' - \mathbf{q}$ and $\mathbf{k}_4 = \mathbf{k} + \mathbf{q}$, respectively. For zero temperature, only those degrees of freedom near $\mu$ become dominant. Due to conservation of momentum in 2D, only the forward scattering is least irrelevant for repulsive interactions in the low-energy window in the renormalization group sense, which consists of two scattering channels: $\mathbf{k}_1 = \mathbf{k}_3$ ($\mathbf{k}_2 = \mathbf{k}_4$) and $\mathbf{k}_1 = \mathbf{k}_4$ ($\mathbf{k}_2 = \mathbf{k}_3$). Then we make a unitary transformation into the chiral basis. For the scattering channel $\mathbf{k}_1 = \mathbf{k}_3$ ($\mathbf{k}_2 = \mathbf{k}_4$), after expansion, there are 16 different combinations in total. The requirement of conservation of energy greatly simplifies the expression, leaving us six terms that read as: $(\hat{k}' \cdot \hat{k})c^\dagger_{k',+}c^\dagger_{k,+}c_{k',+}c_{k,+}$, $-(\hat{k}' \cdot \hat{k})c^\dagger_{k',+}c^\dagger_{k,-}c_{k',+}c_{k,-}$, $(\hat{k}' \cdot \hat{k})c^\dagger_{k',+}c^\dagger_{k,-}c_{k',-}c_{k,+}$, $(\hat{k}' \cdot \hat{k})c^\dagger_{k',-}c^\dagger_{k,+}c_{k',+}c_{k,-}$, $-(\hat{k}' \cdot \hat{k})c^\dagger_{k',-}c^\dagger_{k,+}c_{k',-}c_{k,+}$, $(\hat{k}' \cdot \hat{k})c^\dagger_{k',-}c^\dagger_{k,-}c_{k',-}c_{k,-}$. Only those scattering terms that involve two bands can open a gap in the Dirac state, while the scatterings within one band only renormalize the Fermi velocity but do not contribute to any instabilities. Moreover, the translation invariant mean-field treatment to the second and fifth term contributes to a self-energy that only renormalizes the chemical potential. With these consideration, the forward scattering channel $\mathbf{k}_1 = \mathbf{k}_3$ ($\mathbf{k}_2 = \mathbf{k}_4$) produces the following reduced interaction,

$$H^{(1)}_{eff} = -\frac{V}{4}\sum_{\mathbf{k},\mathbf{k}',n}\left(\hat{k} \cdot \hat{k}'\right)c^\dagger_{\mathbf{k},n}c_{\mathbf{k},-n}c^\dagger_{\mathbf{k}',-n}c_{\mathbf{k}',n}. \tag{16}$$

For the scattering channel $\mathbf{k}_1 = \mathbf{k}_4$ ($\mathbf{k}_2 = \mathbf{k}_3$), following the same approach above, one obtains the following reduced interaction,

$$H^{(2)}_{eff} = -\frac{V}{4}\sum_{\mathbf{k},\mathbf{k}',n}c^\dagger_{\mathbf{k},n}c_{\mathbf{k},-n}c^\dagger_{\mathbf{k}',-n}c_{\mathbf{k}',n}. \tag{17}$$

Both $H^{(1)}_{eff}$ and $H^{(2)}_{eff}$, $p$-wave and $s$-wave type respectively, are reduced from the original Hubbard interaction, which are the least irrelevant channels that can possibly gap out the Dirac cone.

## Data availability
The data that support the findings of this study are available from the corresponding author upon reasonable request.

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

## Acknowledgements
Authors are grateful to A. Burkov, T. Sedrakyan, C.S. Ting, Q.H. Wang, L. Hao, and L.B. Shao for fruitful discussions. This work was supported by 973 Program under Grant No. 2017YFA0303200, and by NSFC (Grants No. 60825402, No. 11574217). O.E. acknowledges the support from ASU startup grant.

## Author contributions
R.W., O.E., B.W. and D.Y.X. performed the calculations, discussed the results and prepared the manuscript.

## Additional information

**Competing interests:** The authors declare no competing interests.

