## [Peer Review File · Nature Communications]

Reviewers' comments:

Reviewer #1 (Remarks to the Author):

Rui Wang et al. studied the effect of strong interaction on the surface states of 3D topological insulator and proposed a new type of topological p+ip excitonic insulator with parity anomaly. They also performed a detailed study on the various properties of this new type of p+ip excitonic insulator, and found that this state can be characterized by a chiral Chern number and also have fractionalized excitations of $e/2$ charge at defects and spontaneous in-plane magnetization. In addition, the manuscript is well written and very clear. I recommend the publication in Nature Communications. Before the final publication, I have the following comments and suggestions:

1. The starting point of the paper is Dirac equation (Eq. 1), which only contain linear terms and is usually to describe the low energy properties of the surfaces states of 3D topological insulator. However, as well known, for many 3D TIs, there are strong warping effect for the Dirac surface states, we need to include the high-order terms. I am wondering whether all the conclusions still hold when we include the high-order terms such as warping effect. It will be better that the authors could add some discussions about this.
2. In addition, there is in fact no particle-hole symmetry for surface states of 3D topological insulator. I am wondering what will happen if we add terms breaking particle-hole symmetry in the Hamiltonian.
3. Although the p+ip excitonic insulator phase is more energetically favored based on the mean-field calculations, but the polar excitonic insulator is also very close. I am wondering whether the mass terms generated from the polar excitonic insulator is negligible. If not, will it affect the conclusions?

Reviewer #2 (Remarks to the Author):

In this paper the authors discuss the realization of the p+ip excitonic insulator on the surface of the topological insulator.

They also studied its general properties such as fractionalization of the Chern number.

While I think the authors' results have some physical importance, I do not see that it suffices the criteria of novelty and impact for Nature Communications.

Actually, the concept of topological p+ip excitonic insulator is not new, and was already pointed out in Hao et al: Phys. Rev. B 82, 195324 (2010), which is not referred in the manuscript.

As for the fractionalization of the Chern number, the authors' argument seems to be a straightforward extension of that of topological p+ip superconductor.

Minor comments:

1) The author's explanation in the second paragraph is confusing, since the s-wave exciton can be spin-singlet as well as spin-triplet in the conventional semiconductors.

It should also be noted that the interaction term given in Eq.(10) does not give rise to the inter-band scattering, nor to the spin-flipping scattering.

Actually, in this interaction, two electrons are scattered from (k,n) to (k',n) and from $(k,-n)$ to $(k',-n)$, respectively, which can be seen as a momentum exchange between an electron and a hole, but does not involve any interband scattering.

2) The derivation from Eq.(9) to (10) should be written more clearly.

3) In Hao et al: Phys. Rev. B 82, 195324 (2010), the possibility of topological p+ip excitonic insulator is studied by solving BCS self-consistent equation for the model with realistic long-range Coulomb interaction.

I do not understand why the authors are not interested in such concrete calculations.

4) The p_x+ip_y excitonic insulator phase might be strongly suppressed or even vanish, if it is treated by some theoretical methods more elaborate than the mean field level.

Reviewer #3 (Remarks to the Author):

In the present manuscript, Wang et al. propose a novel class of excitonic insulators (EIs). EIs discussed in the existing literature almost always assume pairing and condensation of electrons with holes with opposite spins (s-wave EIs, which can be described by BCS formalism). The authors went beyond this tradition and considered the possibility of a p-wave EI in a spin-momentum-locked topological insulator (TI) surface. By developing a detailed theoretical model, they found unique topological properties of such a p+ip EI. They defined an additional chiral Chern number, facilitating to unveil unique properties of the system, such as fractionized electron charge and ground-state spontaneous magnetization. There are also several remarks on candidate materials (like topological Kondo insulators such as SmB₆) that can possibly enable such a physical system. These predictions would be interesting for experimentalists to pursue. The reasoning of the paper is presented in a clear and convincing way, with frequent reference to systems that researchers are more familiar with, such as p+ip superconductors and quantum anomalous Hall systems. These help readers understand the conclusions of the paper not only equation-wise, but also from a big physical picture. Therefore, I believe that this manuscript is potentially suitable for publication in Nature Communications. However, as shown below, I have several questions that I would like the authors to address:

1. The authors should clarify the exciton formation process proposed in Fig. 1. For a Dirac semimetal system (with zero band gap) in Fig. 1(b), if the Fermi level is located at the Dirac point, the thermal energy should be able to create both electron and hole populations at the ground state. I am wondering why, as the authors mentioned, a scattering process is needed to form excitons. This scattering process is depicted as a green arrow representing an interband excitation in Fig. 1. Are the authors proposing to excite excitons through an interband optical absorption process? If so, why is it needed in such a natural Dirac semimetal here?

2. In Fig. 3, the direction of the macroscopic ground-state spontaneous magnetization in k-space is sensitive to the β parameter. However, just by reading the authors' discussions on order parameters, I don't have a direct physical picture of β and how it can be controlled in real systems. This can be important if, as the authors suggested, the system can be probed by anisotropic signatures in transport measurements.

3. The authors should briefly discuss what a polar EI is as a competing phase with the p+ip EI phase (Fig. 4).

4. I suggest that the authors check all notations used throughout the manuscript and state the meanings of several undefined symbols that are not at a first glance obvious to readers.

Reviewer #4 (Remarks to the Author):

he paper studies the excitonic insulator in a surface of strong TI. The main claims include three points. (1) the excitonic insulator has a new type of $p+ip$ order parameter. (2) fractionalized excitations of $e/2$ charge at defects (3) spontaneous in-plane magnetization.

On the whole, this topic is not new, and there are many literatures about the excitonic condensation in the Dirac electron systems. The examples can be found in the references of this paper.

The main claim of point (2) is not surprising, and it is a well-known result in Dirac electron systems. The main claim of point (3) is interesting, but it can be understood with a fundamental rule, which states that the magnetization always spontaneously arises accompanied with the time-reversal symmetry broken. This rule has been well-studied and well-known in the superfluid or superconductor with the pairings breaking TRS.

The most interesting claim is point (1), i.e., the excitonic insulator with intrinsic $p+ip$ pairing. The claim is different from the hybridized system involving the s-wave superconductor or excitonic insulator to induce the effective $p+ip$ type superconductor or excitonic insulator, and also different from a large number of proposals in the references of this paper, in which, the bilayer structure is usually used, and only the s-wave channel is considered.

First, the bilayer has some advantages from the aspect of experiment. The excitons are more stable and have quite long life-time. What is the life-time of the exciton in the single TI surface proposal compared with the bilayer system? Are the excitons fragile or stable? The author should tell the advantage or uniqueness of their proposal to attract the possible attention from the experimenters. Second, in all the bilayer proposal, they do not consider the exciton condensation in the p-wave channel even the p-wave channel is allowed in principle.

The reason lies in that the s-wave channel is more stable than the p-wave channel and has lower energy. In fig.1 (b), the authors argue that the s-wave pairing is forbidden. In my opinion, it is not complete. The authors only consider the electron and hole in pair with same momentum. However, when the electron in the conduction band and the hole in the valence band has inverted momentum, the pairing should be spin-singlet in s-wave channel. Likewise, such s-wave type interaction term is missed from Eq. (9) to (10). I think the probable ground state should still be s-wave type.

The techniques in the paper are standard. All of them can be found in the previous literatures about the excitonic condensation.

the paper is written in oral English in many sentences.

I think this paper is not suited for publication in its present form.

Response to the Reviewers

We thank both of our Referees for their time and consideration in reviewing our paper. Their responses were enthusiastic and helped us to improve our manuscript greatly. Here are our detailed responses to the points and suggestions they have raised:

Reply to referee A

1. Rui Wang et al. studied the effect of strong interaction on the surface states of 3D topological insulator and proposed a new type of topological $p + ip$ excitonic insulator with parity anomaly. They also performed a detailed study on the various properties of this new type of $p + ip$ excitonic insulator, and found that this state can be characterized by a chiral Chern number and also have fractionalized excitations of $e/2$ charge at defects and spontaneous in-plane magnetization. In addition, the manuscript is well written and very clear. I recommend the publication in Nature Communications. Before the final publication, I have the following comments and suggestions.

Response: We are grateful to the referee for the positive evaluation of our paper. The reply to the comments and suggestions are in the following.

2. “The starting point of the paper is Dirac equation (Eq. 1), which only contain linear terms and is usually to describe the low energy properties of the surfaces states of 3D topological insulator. However, as well known, for many 3D TIs, there are strong warping effect for the Dirac surface states, we need to include the high-order terms. I am wondering whether all the conclusions still hold when we include the high-order terms such as warping effect. It will be better that the authors could add some discussions about this.”

Response: We thank the referee for the question. We have followed this suggestion and performed further calculation with taking into account the wrapping effect. With wrapping, the TI surface Hamiltonian can be written as,

$$H_0(\mathbf{k}) = v_F \boldsymbol{\sigma} \cdot \mathbf{k} + \frac{\lambda}{2} (k_+^3 + k_-^3) \sigma_z - \mu, \quad (1)$$

where $k_{\pm} = k_x \pm ik_y$, the wrapping term (the second term above) reduces the continuous rotational symmetry of the Dirac cone state to a discrete C_3 subgroup. μ is the chemical potential, whose effect will be discussed in more detail in the following question. The eigenstate and eigenenergy of Eq.(1) can be readily obtained as,

$$E_{\pm}(\mathbf{k}) = \pm v_F k \sqrt{1 + \frac{\lambda^2}{2v_F^2} k^4 [1 + \cos(6\theta)]} - \mu, \quad (2)$$

and

$$|\mathbf{k}, +\rangle = \left[\frac{1}{\sqrt{2}} e^{-i\theta} x(k, \theta, \lambda), 1 \right]^T, \quad (3)$$

$$|\mathbf{k}, -\rangle = \left[\frac{1}{\sqrt{2}} e^{-i\theta} y(k, \theta, \lambda), 1 \right]^T, \quad (4)$$

where θ is the polar angle of \mathbf{k} and the function $x(k, \theta, \lambda)$, $y(k, \theta, \lambda)$ are defined as,

$$x(k, \theta, \lambda) = \frac{1}{v_F} \left[k^2 \lambda \cos^3 \theta + \frac{1}{\sqrt{2}} \sqrt{\lambda^2 k^4 (1 + \cos(6\theta)) + 2v_F^2 - 3\lambda k^2 \cos \theta \sin^2 \theta} \right], \quad (5)$$

$$y(k, \theta, \lambda) = \frac{1}{v_F} \left[-k^2 \lambda \cos^3 \theta + \frac{1}{\sqrt{2}} \sqrt{\lambda^2 k^4 (1 + \cos(6\theta)) + 2v_F^2 + 3\lambda k^2 \cos \theta \sin^2 \theta} \right]. \quad (6)$$

The eigenvectors above define the unitary transformation that diagonalizes $H_0(\mathbf{k})$ in the chiral basis. Under this unitary transformation, the Hubbard interaction between electrons is transformed correspondingly and is reduced to the following leading vectorial interaction in the forward scattering channel:

$$H_I = \frac{V}{4} \sum_{\mathbf{k}, \mathbf{k}'} e^{i(\theta - \theta')} [xx'^* c_{\mathbf{k}',+}^\dagger c_{\mathbf{k},-}^\dagger - c_{\mathbf{k}',-} c_{\mathbf{k},+} + yy'^* c_{\mathbf{k}',-}^\dagger c_{\mathbf{k},+}^\dagger - c_{\mathbf{k}',+} c_{\mathbf{k},-}], \quad (7)$$

where we have denoted $x(k, \theta, \lambda) = x$, $x(k', \theta', \lambda) = x'$, $y(k, \theta, \lambda) = y$, $y(k', \theta', \lambda) = y'$ for brevity. Note that $e^{i(\theta - \theta')} = \hat{k} \cdot \hat{k}'$. Then, we perform the Hubbard-Stratonovich transformation by introducing the bosonic mean-fields, $\Delta_x = \frac{V}{2} \sum_{\mathbf{k}} x \hat{k} \langle c_{\mathbf{k},-}^\dagger c_{\mathbf{k},+} \rangle$ and $\Delta_y = \frac{V}{2} \sum_{\mathbf{k}} y \hat{k} \langle c_{\mathbf{k},-}^\dagger c_{\mathbf{k},+} \rangle$. This procedure leads to the mean-field Hamiltonian as,

$$H_{MF} = E_+(\mathbf{k})(\tau^z + 1)/2 + E_-(\mathbf{k})(1 - \tau^z)/2 - \tau^+(x\Delta_x \cdot \hat{k} + y\Delta_y \cdot \hat{k})/2 - \tau^-(x\Delta_x^* \cdot \hat{k} + y\Delta_y^* \cdot \hat{k})/2. \quad (8)$$

We solved the self-consistent mean-field equations for both the chiral p+ip excitonic insulator (CEI) and the polar excitonic insulator (PEI) cases and still find the CEI is more stable. For the CEI orders, we have $\Delta_{x,y} = \Delta_{x,y}(\mathbf{e}_x - i\mathbf{e}_y)/\sqrt{2}$. The mean-field equations can be derived by variation of the total ground state energy with respect to the order parameters, i.e., $\delta E_G(\lambda)/\delta \Delta_x = 0$ and $\delta E_G(\lambda)/\delta \Delta_y = 0$.

For zero wrapping with $\lambda = 0$, we return back to the results in the manuscript with $\Delta_x = \Delta_y = \Delta_c$. For nonzero λ , we calculate the self-consistent order parameters from the above mean-field equations and plot $\Delta_{x,y}(\lambda)/\Delta(\lambda = 0)$ as a function of wrapping λ . As shown by Fig.1R, we find that $\Delta_x \simeq \Delta_y$, giving rise to a single uniform exciton gap, more importantly, even though the exciton gap $\Delta(\lambda)$ gradually decreases with increasing λ , it maintains more than 75 percent for very strong wrapping with $\lambda = \hbar v_F k$.

Fig1R: The exciton gap $\Delta(\lambda)$ as a function of the wrapping λ (normalized by $\Delta(\lambda = 0)$). The x-axis is in unit of the energy cutoff of the surface state $v_F \hbar \Lambda$.

Therefore, we conclude that the p -wave exciton phase is not fragile but quite stable against higher order wrapping of the TI surface. For small wrapping, it can be safely regarded as weak perturbation and all the discussed properties of the topological excitonic insulator still hold. We have followed the referee's suggestion and added some additional discussions on the wrapping effect in the revised manuscript.

2. “ In addition, there is in fact no particle-hole symmetry for surface states of 3D topological insulator. I am wondering what will happen if we add terms breaking particle-hole symmetry in the Hamiltonian. ”

Response: We thank the referee for this important question. We have performed further analysis with taking into account the particle-hole symmetry (PHS) breaking term. The simplest term that breaks the PHS is the finite chemical potential on the TI surface. Let us consider the effect of μ for example. With $\mu \neq 0$, the eigenenergy of the mean-field Hamiltonian is $E_{\pm} = -\mu \pm \sqrt{v_F^2 k^2 + |\mathbf{\Delta} \cdot \mathbf{k}|^2}$, from which the zero temperature mean-field equation can be derived as,

$$\frac{1}{V} = \frac{1}{4} \sum_{\mathbf{k} \in \text{occ}} \frac{|\mathbf{\Delta} \cdot \mathbf{k}|^2}{\sqrt{v_F^2 k^2 + |\mathbf{\Delta} \cdot \mathbf{k}|^2}}, \quad (9)$$

where the sum of \mathbf{k} is over all the occupied states. From the above mean-field equation, we know that the effect of chemical potential μ is only involved in the sum of \mathbf{k} . For $\mu < \Delta$, all the states in the valence band are filled while those in the conduction band are empty. In this case, the above mean-field theory remains the same as the $\mu = 0$ case, therefore all the conclusions in our manuscript remain correct. However, when $\mu > \Delta$, the sum of momentum states not only includes the valence band but also the filled states in the conduction band, this will modify the mean-field result quantitatively. Specifically, with a finite μ , one can obtain the critical interaction V_c as

$$\frac{V_c}{2v_F\Lambda} = \frac{v_F\Lambda}{v_F\Lambda - \mu}. \quad (10)$$

V_c is the threshold for the interaction above which the exciton insulator can be formed. We see that for $\mu = 0$, $V_c = 2v_F\Lambda$, reducing back to the V_c in the manuscript. Whereas, a larger μ increases the critical interaction. Therefore, even though the exciton insulator is more difficult to form, it will always be stable if the interaction is larger than V_c . When this condition is satisfied, all the conclusions in the particle-hole symmetric case will not be unchanged by the PHS breaking term.

From above, we know that the smaller chemical potential will be more beneficial to exciton instability. In practice, there are experimental methods to tune the chemical potential of the surface state, such as using a gating voltage (Phys. Rev. Lett. 105, 176602) or introducing dopants (Phys. Rev. Lett. 106, 196801). It will be more favorable if the low-energy physics is dominated by those degrees of freedom that are close to the Dirac point. Then, the exciton insulator can be spontaneously generated with a strong interaction.

3. Although the p+ip excitonic insulator phase is more energetically favored based on the mean-field calculations, but the polar excitonic insulator is also very close. I am wondering whether the mass terms generated from the polar excitonic insulator is negligible. If not, will it affect the conclusions?”

Our response: To answer this question, we have performed an extended mean-field study which treats the polar excitonic insulator (PEI) and chiral $p + ip$ excitonic insulator (CEI) states on equal footing. Specifically, we decouple the interaction into both channels and introduce the PEI and CEI mean-field orders Δ_c and Δ_p simultaneously. This leads to the mean-field Hamiltonian with two excitonic order parameters as,

$$H_{mf} = v_F k \tau_z - \Delta_p \cos \theta \tau^x - \frac{1}{\sqrt{2}} \Delta_c e^{-i\theta} \tau^+ - \frac{1}{\sqrt{2}} \Delta_c e^{i\theta} \tau^-, \quad (11)$$

together with constant terms proportional to Δ_c^2 and Δ_p^2 which are not shown. By minimizing the ground state free energy with respect to both Δ_p and Δ_c , one can determine the coupled self-consistent equations, based on which one can determine whether both Δ_c and Δ_p coexist or not. Interestingly, we find that, for

any interaction $V > V_c$, the final converged solution only has $\Delta_c \neq 0$. The CEI order Δ_c will completely suppress the PEI order Δ_p and results in $\Delta_p = 0$ in the final ground state.

Therefore, it is known that even though the two states have close mean-field energy, only the $p + ip$ order rather than the polar order can be the true instability. The competition from the polar order becomes irrelevant in the ground state. So, the mass term generated from polar excitonic insulator can be neglected at least in mean-field level.

The referee's question also motivates us to study the thermal fluctuation as well as the quantum fluctuation beyond the mean-field saddle point. We found that a Kosterlitz-Thouless transition will take place and the emergence of the energy scale T_{KT} that describes the confinement of vortices at finite temperature. However, this has already gone beyond the scope of the current study. We will present our results on the fluctuations in a subsequent work.

Reply to referee B

1. “In this paper the authors discuss the realization of the $p + ip$ excitonic insulator on the surface of the topological insulator. They also studied its general properties such as fractionalization of the Chern number. While I think the authors’ results have some physical importance, I do not see that it suffices the criteria of novelty and impact for Nature Communications. Actually, the concept of topological $p+ip$ excitonic insulator is not new, and was already pointed out in Hao et al: Phys. Rev. B 82, 195324 (2010), which is not referred in the manuscript. As for the fractionalization of the Chern number, the authors’ argument seems to be a straightforward extension of that of topological $p+ip$ superconductor.”

Response: We disagree with the Referee. Even though the concept of topological excitonic insulator was indeed proposed by earlier works such as Phys. Rev. B 82, 195324 (2010), where a combination of semiconductor heterostructure with two 2D electron gas (2DEG), ferromagnetic layers and Rashba spin-orbit coupling is required; we put forward a minimal topological excitonic insulator model for the first time, where the $p+ip$ symmetry is intrinsically realized by spontaneous symmetry breaking. The minimal model enjoys fractional chiral Chern $C_c = 1/2$ rather than the integer (Thouless, Kohmoto, Nightingale, and Nijs) TKNN number in earlier concepts, and therefore it is the more elemental theory which can be used for construction of more complicated models for topological excitonic insulators. More interestingly, we show that this minimal model can be inherently realized on the topological insulator surface state, enriching the understanding of strong correlation physics on topological materials. Besides, our work also has some technique advances, e.g., we have demonstrated that the topological behavior is originated from the parity anomaly, which induces a Chern-Simons gauge field theory to a vortex state, resulting in the charge fractionalization. To the best of our knowledge, the relation between parity anomaly and the topological nature of the $p + ip$ wave excitonic insulator has not been discussed in previous literatures. Even though the underlying mathematics is similar to the topological $p+ip$ superconductors, however the physical setup is completely different and therefore leads to completely different physical properties such as the anisotropic excitonic current discussed in the manuscript. Moreover, in our revised manuscript, we have further discussed a new promising way to further stabilize the predicted $p+ip$ excitonic insulator, i.e., the TI surface driven by linearly-polarized external field. We found that the interband optical excitations renormalizes the TI surface and generates an effective p-wave interaction between electrons, favoring the proposed $p + ip$ excitonic instability. Therefore, our work is not only different from earlier works, but also has new findings and original ideas that merit publication on Nature Communications.

2. “The author’s explanation in the second paragraph is confusing, since the s-wave exciton can be spin-singlet as well as spin-triplet in the conventional semiconductors. It should also be noted that the interaction term given in Eq.(10) does not give rise to the inter-band scattering, nor to the spin-flipping scattering. Actually, in this interaction, two electrons are scattered from $(k'n)$ to (k,n) and from $(k,-n)$ to $(k',-n)$, respectively, which can be seen as a momentum exchange between an electron and a hole, but does not involve any interband scattering.”

Response: We thank the referee for the question which motivates our additional analysis on the exciton formation. Generally speaking, for excitons in conventional semiconductors, the spin of electron and hole can indeed be either parallel or anti-parallel. However, when a many-body system of excitons in traditional semiconductor is considered, the s-wave singlet type condensation is usually more energetically favorable. The referee’s question, to our understanding, includes two-fold aspects. One is about Fig.1 and the discussion in the second paragraph, the other is about the Eq.(10) and how it is related to Fig.1.

In terms of the first aspect, we would like to clarify Fig.1 and the discussion in the second paragraph.

Here, we are actually discussing the interband optical absorption process. In this particular case, the excited excitons form spin-singlet rather than triplet for the conventional semiconductor with no spin-orbit coupling, as will be clear in the following: We consider an external light field described by the vector potential field \mathbf{A} . Then, the perturbation on the system H_0 is formulated as (Phys. Rev. B 88, 075144 (2013)), $H' = -e \frac{\partial H_0}{\partial \mathbf{k}} \cdot \mathbf{A}$. Let us consider a linearly-polarized light that is perpendicularly applied onto the 2D system, i.e., $\mathbf{A} = A_x \mathbf{e}_x + A_y \mathbf{e}_y$ where $A_{x/y} = |A_{x/y}| \cos \omega t$. For TI surface with $H_0 = v_F \sigma \cdot \mathbf{k}$, it is clear that $H'_{TI} = -ev_F(A_x \sigma_x + A_y \sigma_y)$. While for a conventional semiconductor with $H_0 = \xi_{\mathbf{k}} \sigma^0 \tau_z$ where $\xi_{\mathbf{k}} = \sqrt{(k^2/2m)^2 + \Delta^2}$, the unity matrix σ^0 occurs due to the spin degeneracy and τ denotes the band degrees of freedom, we have $H'_{Semi} = -e \nabla_{\mathbf{k}} \xi_{\mathbf{k}} \sigma^0 \tau_z \cdot \mathbf{A}$. Comparing between H'_{TI} and H'_{Semi} , we know that there are spin flipping matrices σ_x, σ_y in the former while only the spin conserving matrix σ^0 in the latter. This means that, the interband direct excitation is accompanied by spin-flipping in TI surface which is absent in conventional semiconductors. Therefore the excited excitons, if they are stable, enjoy different spin features in the two systems, as indicated by Fig.1 in the manuscript. This point was not sufficiently demonstrated in the previous manuscript, we have made a lot of improvements in the revised version.

As to the second aspect, we agree with the referee's comment on the interaction term Eq.(10). However, we are not asserting that this interaction introduces spin-flipping or interband scattering, and are not relating Eq.(10) to Fig.1, which discusses the interband optical processes. We are just demonstrating that with the interaction Eq.(10) (which involves electrons within both bands, since we have both n and $-n$ electrons), a $p + ip$ excitonic insulator is the most stable ground state on TI surface. In our previous manuscript, we did not demonstrate sufficiently how the interband optical processes are needed for realization of $p + ip$ excitonic insulator. In the revised manuscript, we have further presented the study of a linearly-polarized light. The electron-photon coupling and the interband optical processes are found to be able to drive the system into the predicted $p + ip$ excitonic insulator, verifying the mechanism for the spin-triplet excitons illustrated by Fig.1.

3. “ The derivation from Eq.(9) to (10) should be written more clearly.”

Response: We have followed the suggestion and added detailed derivation from Eq.(9) to (10) in the Methods.

4 “ In Hao et al: Phys. Rev. B 82, 195324 (2010), the possibility of topological p+ip excitonic insulator is studied by solving BCS self-consistent equation for the model with realistic long-range Coulomb interaction. I do not understand why the authors are not interested in such concrete calculations.”

Our response: In our previous manuscript, our motivation was to give the most compact model to describe the main physics therefore we only considered the short-range Hubbard interactions. Here, following the suggestion of the referee we study a long-range Coulomb interaction at finite temperature. We also consider a more general case where there is an electron and a hole pocket on the conduction and valence band of TI surface, with chemical potential μ^+ and μ^- respectively. Our previous model is the limit case with $\mu^+ = \mu^- = 0$. We study the interaction with screening wave vector κ ,

$$H_{int} = \sum_{\mathbf{k}, \mathbf{k}', \mathbf{q}} \frac{2\pi e^2}{(\kappa + q)\epsilon} c_{\mathbf{k}+\mathbf{q}\sigma}^\dagger c_{\mathbf{k}'-\mathbf{q}\sigma'}^\dagger c_{\mathbf{k}'\sigma'} c_{\mathbf{k}\sigma} \quad (12)$$

where $\kappa = 2\pi e^2(N_0^+ + N_0^-)/\epsilon = \alpha(k_F^+ + k_F^-)$ with $\alpha = e^2/\epsilon v_F \hbar$, k_F^\pm being the Fermi wave vector of the electron and hole pocket respectively. Using the same method as in the manuscript, one can show the interaction consists of the following long-range p -wave interaction component after restricting to the

forward scattering channel, i.e.,

$$H_{int} = - \sum_{\mathbf{k}, \mathbf{k}'} \frac{2\pi e^2}{(\kappa + |\mathbf{k} - \mathbf{k}'|)\epsilon} c_{\mathbf{k}'\sigma}^\dagger c_{\mathbf{k}'\bar{\sigma}} c_{\mathbf{k}\bar{\sigma}}^\dagger c_{\mathbf{k}\sigma}. \quad (13)$$

where the minus sign comes from the exchange of fermion operators. After making the unitary transformation to the band basis, the interaction reads,

$$H_{int} = - \sum_{\mathbf{k}, \mathbf{k}'} \frac{2\pi e^2 \hat{\mathbf{k}} \cdot \hat{\mathbf{k}'}}{(\kappa + |\mathbf{k} - \mathbf{k}'|)\epsilon} c_{\mathbf{k}'+}^\dagger c_{\mathbf{k}'-} c_{\mathbf{k}-}^\dagger c_{\mathbf{k}+}. \quad (14)$$

In the following, we introduce $V_{\mathbf{k}-\mathbf{k}'} = \frac{2\pi e^2}{(\kappa + |\mathbf{k}-\mathbf{k}'|)}$, so that

$$H_{int} = - \sum_{\mathbf{k}, \mathbf{k}'} V_{\mathbf{k}-\mathbf{k}'} \hat{\mathbf{k}} \cdot \hat{\mathbf{k}'} c_{\mathbf{k}'+}^\dagger c_{\mathbf{k}'-} c_{\mathbf{k}-}^\dagger c_{\mathbf{k}+}. \quad (15)$$

The noninteracting Hamiltonian H_0 has an electron and a hole pocket: $(v_F k - \mu^+) c_{\mathbf{k}+}^\dagger c_{\mathbf{k}+}$ for the electrons living in the conduction band, and $(-v_F k - \mu^-) c_{\mathbf{k}-}^\dagger c_{\mathbf{k}-}$ living in the valence band. By introducing Hubbard-Stratonovich decomposition in terms of H_{int} and neglecting the fluctuation of the bosonic field $\Delta_{\mathbf{k}} = \sum_{\mathbf{k}'} V_{\mathbf{k}-\mathbf{k}'} \langle \hat{\mathbf{k}} c_{\mathbf{k}'-}^\dagger c_{\mathbf{k}'+} \rangle$, the interaction is decoupled as,

$$H_{int} = - \sum_{\mathbf{k}} \Delta_{\mathbf{k}} \cdot \hat{\mathbf{k}} c_{\mathbf{k}-}^\dagger c_{\mathbf{k}+} - \sum_{\mathbf{k}} \Delta_{\mathbf{k}}^* \cdot \hat{\mathbf{k}} c_{\mathbf{k}+}^\dagger c_{\mathbf{k}-} + \sum_{\mathbf{k}\mathbf{k}'} V_{\mathbf{k}-\mathbf{k}'}^{-1} \Delta_{\mathbf{k}} \cdot \Delta_{\mathbf{k}'}^*, \quad (16)$$

leading to the total mean-field Hamiltonian

$$H_{MF} = \sum_{\mathbf{k}} \begin{pmatrix} c_{\mathbf{k},+} \\ c_{\mathbf{k},-} \end{pmatrix}^\dagger \begin{pmatrix} v_F k - \mu^+ & -\Delta_{\mathbf{k}} \cdot \hat{\mathbf{k}} \\ -\Delta_{\mathbf{k}}^* \cdot \hat{\mathbf{k}} & -v_F k - \mu^- \end{pmatrix} \begin{pmatrix} c_{\mathbf{k},+} \\ c_{\mathbf{k},-} \end{pmatrix} + \sum_{\mathbf{k}\mathbf{k}'} V_{\mathbf{k}-\mathbf{k}'}^{-1} \Delta_{\mathbf{k}} \cdot \Delta_{\mathbf{k}'}^*, \quad (17)$$

The mean-field self-consistent equation can be derived in standard way by minimizing the ground state energy. For finite temperature, after inserting the chiral excitonic order parameter with $\Delta_{\mathbf{k}} = \frac{1}{\sqrt{2}} \Delta_{\mathbf{k}}^c (\hat{\mathbf{e}}_x - i\hat{\mathbf{e}}_y)$, the self-consistent equation is obtained as,

$$\Delta_{\mathbf{k}} = \frac{1}{2} \int_{\mathbf{k}' \in \Omega} \frac{d\mathbf{k}'}{(2\pi)^2} V_{\mathbf{k}-\mathbf{k}'} \Delta_{\mathbf{k}'} \frac{n_F(\xi_{\mathbf{k}} - \mu_1) - n_F(-\xi_{\mathbf{k}} - \mu_1)}{\xi_{\mathbf{k}}}, \quad (18)$$

with

$$\xi_{\mathbf{k}} = \sqrt{(v_F k - \mu_2)^2 + |\Delta_{\mathbf{k}}^c|^2/2}, \quad (19)$$

and we have defined $\mu_1 = (\mu^+ + \mu^-)/2$ and $\mu_2 = (\mu^+ - \mu^-)/2$. In Eq.(18), the integral of 2D momentum \mathbf{k} is performed within an effective cutoff Λ dependent on materials (for Be_2Si_3 , $\Lambda = 0.3$ eV), n_F is the Fermi distribution function. From the above mean-field theory, we can determine the T_c as a function of μ_1 , μ_2 , and we can determine the exciton gap $\Delta_{\mathbf{k}}^c$.

The phase diagram as a function of T and μ_2 is obtained in Fig.2R.

Fig.2R: The mean-field phase diagram of TI surface with a strong long-range Coulomb (screened) interaction between electrons, as a function of chemical potential μ_2 and temperature T . The parameters used are: $\alpha = e^2/\epsilon v_F = 0.6$, $\hbar v_F \Lambda = 0.3eV$, $v_F = 3 \times 10^5$ m/s.

We see that, for a long-range interaction, the excitonic order still occurs for low temperature regime, above which, the system remains a particle-hole gas (liquid). We also make extension by going beyond the saddle point solution of mean-field equations. With the thermal fluctuation taken into account, we find the first order transition is replaced by a Kosterlitz-Thouless type transition. The calculated KT transition temperature is denoted by the brown marks in the above phase diagram, which is a little lower than the mean-field estimation of T_c .

We have added some discussion in the revised paper on the long-range interaction. However, we would like to present all the above details in a more comprehensive subsequent study.

4. “ The $p_x + ip_y$ excitonic insulator phase might be strongly suppressed or even vanish, if it is treated by some theoretical methods more elaborate than the mean field level.”

Response: The strong interaction on Dirac state becomes an open question when one goes beyond mean-field. In a very recent work (Science 361, 570 (2018)), the authors have numerically verified that a gapped Mott insulating state takes place on the 2D Dirac state (graphene) when the electron-electron interaction is strong enough. It is interesting to apply heavy numerics to our model and to testify the mean-field theory and see if the Mott insulating state corresponds to an excitonic order, however, this goes beyond the scope of the current work. Nevertheless, we still followed the referee’s suggestion and considered the thermal fluctuation around the mean-field saddle point. We found that the $p + ip$ wave exciton insulator can still be the leading instability as a result of interaction Eq.(10) in the manuscript. Moreover, vortex solutions are found to be more stable than the uniform mean-field, and the Kosterlitz-Thouless transition takes place at the temperature T_{KT} . We have calculated T_{KT} based on the self-consistent condition on the superfluid density $T_{KT} = \frac{\pi}{2} \rho_s(T_{KT})$ (Phys. Rev. B 78, 121401(R)). The obtained T_{KT} is denoted by the brown marks in the above figure, which can reach up to 40K for a strong long-range interaction

$\alpha = e^2/\epsilon v_F = 0.6$ (with other parameters taken from Bi_2Se_2 , i.e., $v_F = 3 \times 10^5 m/s$ and the energy cutoff $\hbar v_F \Lambda = 0.3 eV$). Below T_{KT} , the $p + ip$ -wave excitonic condensation forms.

In short, with taking into account the fluctuation around the mean-field solution, the excitonic insulator is still found to be stable. This should partly clarify the referee's concern. We still do not know the exact answer if one completely goes beyond the mean-field theory. This should be an interesting open question to be revealed by numerical methods (such as diagrammatic quantum Monte Carlo) in the future.

Reply to referee C

1. “In the present manuscript, Wang et al. propose a novel class of excitonic insulators (EIs). EIs discussed in the existing literature almost always assume pairing and condensation of electrons with holes with opposite spins (s-wave EIs, which can be described by BCS formalism). The authors went beyond this tradition and considered the possibility of a p-wave EI in a spin-momentum-locked topological insulator (TI) surface. By developing a detailed theoretical model, they found unique topological properties of such a p+ip EI. They defined an additional chiral Chern number, facilitating to unveil unique properties of the system, such as fractionized electron charge and ground-state spontaneous magnetization. There are also several remarks on candidate materials (like topological Kondo insulators such as SmB6) that can possibly enable such a physical system. These predictions would be interesting for experimentalists to pursue. The reasoning of the paper is presented in a clear and convincing way, with frequent reference to systems that researchers are more familiar with, such as p+ip superconductors and quantum anomalous Hall systems. These help readers understand the conclusions of the paper not only equation-wise, but also from a big physical picture. Therefore, I believe that this manuscript is potentially suitable for publication in Nature Communications. However, as shown below, I have several questions that I would like the authors to address.”

Response: We are grateful to the referee for the positive evaluation of our work. The questions have been carefully considered and addressed in the revised paper. Our brief reply is in what follows.

2. “The authors should clarify the exciton formation process proposed in Fig. 1. For a Dirac semimetal system (with zero band gap) in Fig. 1(b), if the Fermi level is located at the Dirac point, the thermal energy should be able to create both electron and hole populations at the ground state. I am wondering why, as the authors mentioned, a scattering process is needed to form excitons. This scattering process is depicted as a green arrow representing an interband excitation in Fig. 1. Are the authors proposing to excite excitons through an interband optical absorption process? If so, why is it needed in such a natural Dirac semimetal here?”

Response: We thank the referee for this question. In our previous manuscript, this point was indeed not sufficiently discussed. In the revised manuscript, we have added a thorough illustration on Fig.1 and on its relevance to the mean-field model. We also input some additional content that clarify how Fig.1 is related to our theory. Our brief answer is in what follows.

First, in Fig.1, we are indeed comparing the differences between TI surface and the conventional semiconductors in terms of the interband optical process. One can show that for the TI surface, the effect of linearly-polarized light, $\mathbf{A} = A_x \mathbf{e}_x + A_y \mathbf{e}_y$, is to introduce the perturbation term: $H'_{TI} = -ev_F(A_x \sigma_x + A_y \sigma_y)$. Whereas, for a conventional semiconductor with $H_0 = \xi_{\mathbf{k}} \sigma^0 \tau_z$ where $\xi_{\mathbf{k}} = \sqrt{(k^2/2m)^2 + \Delta^2}$, the unity matrix σ^0 occurs due to the spin degeneracy and τ denotes the band degrees of freedom, we have $H'_{Semicon} = -e\nabla_{\mathbf{k}} \xi_{\mathbf{k}} \sigma^0 \tau_z$. Comparing between H'_{TI} and $H'_{Semicon}$, we know that there are spin flipping matrices σ_x, σ_y in the former while only the spin conserving matrix σ^0 in the latter. This means that, the interband excitation is accompanied by spin-flipping in TI surface which is absent in conventional semiconductors. This is a key difference we pointed out by Fig.1 before going to details of mean-field models.

Then, we discussed the mean-field theory with the specific case where the Fermi energy is located at the Dirac point. In this case, the thermal excitation generates both electron and hole states and excitonic instability is shown to take place without any optical processes, exactly as the referee pointed out. We

proved that the TI surface has a strong tendency towards the $p+ip$ excitonic instability with a significant parameter region stabilizing this phase in the phase diagram, however, the s -wave state is still more energetically stable for the local Hubbard model (In our previous manuscript, we did not make explicit comparison between the energy of the s -wave state and the $p+ip$ state. A more complete mean-field theory is presented in the revised manuscript which gives the full phase diagram). In order to further stabilize the $p+ip$ excitonic insulator, we propose to utilize the interband optical processes illustrated by Fig.1. In the revised manuscript, we further prove that additional photons and the optical processes depicted by Fig.1 play important role and can renormalize the Dirac surface state by generating an effective p -wave-type interaction, resulting in a true $p+ip$ excitonic insulator ground state. This verifies our physical picture given by Fig.1 at the beginning of the paper. This additional study should have clarified the referee's concern on how the theory is related to the interband optical processes in Fig.1, i.e., they are needed for stabilization of the $p+ip$ excitonic order from the s -wave phase.

We again thank the Referee for the comment and we now hope that this point is more clear.

3. “In Fig. 3, the direction of the macroscopic ground-state spontaneous magnetization in k -space is sensitive to the β parameter. However, just by reading the authors discussions on order parameters, I don't have a direct physical picture of β and how it can be controlled in real systems. This can be important if, as the authors suggested, the system can be probed by anisotropic signatures in transport measurements.”

Response: β is the global phase associated with the excitonic order parameter, i.e., $\Delta_c e^{i\beta}(k_x - ik_y)$. Since β is \mathbf{k} -independent, it is redundant with regards to energetics, i.e., different β states are degenerate in energy. We have shown that β is also associated with the direction of the in-plane magnetization of the exciton phase. In physical picture, one can understand the quantity β through the analogy with the superconductor phase θ in $\Delta e^{i\theta}$. Superconductors (SCs) with only different θ 's are degenerate in energy with each other, however, θ can manifest itself by the Josephson current in SC junction that has a phase difference $\Delta\theta$. Here, the phase β can also be manifested by such junctions, because the magnetic domain wall will be generated at the boundary between two regions with a phase difference $\Delta\beta$.

We note that, similar to the SC phase θ , the phase β is randomly picked during the process of symmetry breaking. This is again similar to the ferromagnetic orders, which, after cooling, can develop random direction of magnetization (because all directions are degenerate in energy). However, recalling that if one introduces a small magnetic field along a certain direction \mathbf{n} while cooling, the system will have a large probability to develop a magnetization along \mathbf{n} . The situation is the same in our predicted $p+ip$ exciton insulator, which can be regarded as an in-plane ferromagnetic (but topological) order formed by electrons. Therefore, we expect to control the value of β by applying a small in-plane magnetic field (small enough to be a weak perturbation on the TI surface) while lowering the temperature.

We have followed the referee's suggestion and added more discussions on this point.

4. “The authors should briefly discuss what a polar EI is as a competing phase with the $p+ip$ EI phase (Fig. 4). ”

Response: The polar EI is also a coherent pairing state of electron and holes (on TI surface). The difference between the polar EI and the chiral $p+ip$ EI lies in the pairing symmetry. For chiral $p+ip$ EI, the pairing symmetry is $p+ip$ wave and hence is topological, while for polar EI, the pairing is just normal p -wave with its order parameter $\Delta_p \cos\theta$. Similar to the 2D superconductor case, the normal p -wave order here is topological trivial and has zero chiral Chern number $C_c = 0$. We have discussed more about the nature of the polar EI in the revised paper.

5. “I suggest that the authors check all notations used throughout the manuscript and state

the meanings of several undefined symbols that are not at a first glance obvious to readers.
”

Our response: We have carefully checked all the notations, corrected typos, and explained the undefined symbols.

Reply to referee D

1. “The paper studies the excitonic insulator in a surface of strong TI. The main claims include three points. (1) the excitonic insulator has a new type of p+ip order parameter. (2) fractionalized excitations of $e/2$ charge at defects (3) spontaneous in-plane magnetization. On the whole, this topic is not new, and there are many literatures about the excitonic condensation in the Dirac electron systems. The examples can be found in the references of this paper. The main claim of point (2) is not surprising, and it is a well-known result in Dirac electron systems. The main claim of point (3) is interesting, but it can be understood with a fundamental rule, which states that the magnetization always spontaneously arises accompanied with the time-reversal symmetry broken. This rule has been well-studied and well-known in the superfluid or superconductor with the pairings breaking TRS. The most interesting claim is point (1), i.e., the excitonic insulator with intrinsic p+ip pairing. The claim is different from the hybridized system involving the s-wave superconductor or excitonic insulator to induce the effective p+ip type superconductor or excitonic insulator, and also different from a large number of proposals in the references of this paper, in which, the bilayer structure is usually used, and only the s-wave channel is considered.”

Response: We thank the referee for the comment. Our proposal is indeed very different from the earlier literatures that usually focus on bilayer, s-wave condensation channel cases. Our answer to the referee’s questions is in what follows.

2. “First, the bilayer has some advantages from the aspect of experiment. The excitons are more stable and have quite long life-time. What is the lift-time of the exciton in the single TI surface proposal in compared with the bilayer system? Are the excitons fragile or stable? The author should tell the advantage or uniqueness of their proposal to attract the possible attention from the experimenters. ”

Response: We agree with the referee that this question needs to be addressed. The reason why bilayer system has attracted large attention for studying exciton condensation is that the electron and hole is spatially separated and with an electric dipole, which significantly enlarges the lifetime compared to conventional semiconductors. In TI systems for bulk metallic samples, earlier experimental works have reported the exciton lifetime to be a few picoseconds (Phys. Rev. Lett. 108, 117403 (2012), Phys. Rev. Lett. 109, 127401 (2012), Nat. Commun. 5, 3003 (2014)). Remarkably, the later works have observed a gigantic lifetime exceeding $4\mu s$ (Phys. Rev. Lett. 115, 116801 (2015)). This remarkably long lifetime was only observed when the chemical potential lies at the surface state, where the bulk bands are not excited as the pump light is shone. The proper understanding of such gigantic lifetime is still absent nowadays, however, it was argued to be attributed to the spin-momentum locking of the surface state (Phys. Rev. Lett. 115, 116801 (2015)). In general the exciton lifetime in TI (or TI surface) is from a few picoseconds to a few microseconds, significantly larger than than that in graphene. These finding demonstrate that TI surface can be a rare platform to study (quasi-equilibrium) excitonic physics.

Moreover, we would like to share with the referee some information we acquired by collaborating with an experimental group recently (the work has been submitted for review, please see our preprint, arXiv: 1810.10653 (2018)). In our sample (Sb-doped Bi_2Se_3), the measured lifetime is up to 20 ns. Although not perfectly long, the system can reach to a quasi-equilibrium state around 100 fs after the pumping excitation (Nano Lett. 8, 4248 (2008), Phys. Rev. Lett. 108, 167401 (2012), Nat. Mater. 12, 1119 (2013)). By keep shining the laser onto the sample, the experimentalists observe a long-range photocurrent that beyond any explanation based on free carrier models (arXiv: 1810.10653 (2018)). This indicates the formation of a quasi-equilibrium excitonic condensation superfluid-like state, verifying our theoretical prediction of excitonic instabilities on the TI surface.

Besides, another advantage of our proposal is that the long-range Coulomb interaction could be more effective in TI surface to generate excitonic order. Here, the ratio between Coulomb energy and the electron kinetic energy $\alpha = E_C/E_K$ is formulated as $\alpha = e^2/(4\pi\epsilon\hbar v_F\nu)$ due to the linear dispersion of the TI surface state, where e is the electron charge, ϵ is the dielectric constant of the material, v_F is the Fermi velocity, ν is the degeneracy of Dirac cones where $\nu = 1$ for TI surface. The surface state with a single non-degenerate Dirac cone has the lowest ν , a relatively low v_F (compared to graphene), the reduced ϵ at surface can be tuned even lower by doping or gating (Nat. Phys. 7, 939 (2011)). This can lead to a large α in favor of the exciton condensates (For the calculation of the long-range interaction model, please see our reply to 4-th question of referee B).

According to the above experimental literatures as well as our own experimental data, the excitons in TI surface are much more stable (the longest lifetime measured by experiment so far reaches to $4\mu s$) than those in conventional semiconductors and can indeed stimulate great interest in terms of the excitonic condensation in the future. Following the suggestion of the referee, we have input more discussions on this aspect in the section ‘‘Discussion’’.

3. ‘‘Second, in all the bilayer proposal, they do not consider the exciton condensation in the p-wave channel even the p-wave channel is allowed in principle. The reason lies in that the s-wave channel is more stable than the p-wave channel and has lower energy. In fig.1 (b), the authors argue that the s-wave pairing is forbidden. In my opinion, it is not complete. The authors only consider the electron and hole in pair with same momentum. However, when the electron in the conduction band and the hole in the valence band has inversed momentum, the pairing should be spin-singlet in s-wave channel. Likewise, such s-wave type interaction term is missed from Eq. (9) to (10). I think the probable ground state should still be s-wave type.’’

Response: We totally understand the referee’s concern. Usually, the s-wave channel has lower excitonic condensation energy, such as in bilayer systems. Therefore, we need to clarify how the p-wave condensation can be more stable in our proposal. In our previous manuscript, this point was indeed not sufficiently demonstrated. We only intended to show that a p -wave interaction channel automatically arises from the local Hubbard interaction, as a result of the spin-momentum locking of TI surface. Therefore, the system has a strong tendency to form the $p + ip$ topological exciton insulator. More results are now presented to show how to realize the true $p + ip$ excitonic insulator in the revised manuscript. Besides, we would like to clarify Fig.1(b), where we are discussing the interband optical process on TI surfaces. We consider the participation of a photon. It will generate *direct* interband excitation, forming a pair of electron and hole with same \mathbf{k} (and the same spins). This indicates that the interband optical process is in favor of the p -wave instability. Following the referee’s suggestion, we have performed further calculations and clarified corresponding issues. The reply is in what follows.

First, starting from the Hubbard interaction, as the referee suggests, one cannot excludes the s -wave instability. We agree with this point. A complete mean-field calculation treating the s -wave and p -wave channel on equal footing are presented in the revised manuscript. The phase diagram is obtained and shown in Fig.3R.

Fig.3R: The mean-field phase diagram showing excitonic instabilities of TI surface with local Hubbard model. We treat V_p and V_s as two independent interaction parameters.

As shown, we indeed find a $p+ip$ excitonic insulator region in the phase diagram, in addition to the s -wave excitonic insulator. This is a key difference from conventional semiconductors or the bilayer systems where no $p+ip$ region exists (because the absence of any V_p channel). This difference is due to the additional p -wave interaction channel V_p emerging due to the spin-momentum locking of the TI surface. However, for a specific local Hubbard interaction, we have proved that the p -wave and s -wave interaction component has equal strength $V_s = V_p$ (see the Methods in the revised manuscript or below). This makes the s -wave excitonic insulator to be more stable than the $p+ip$ -wave phase for the discussed Hubbard interaction (the red dashed line only enters into the s -wave region), verifying the intuitive suggestion by the referee.

However, the story does not end here. Up to now, we only discussed the natural Dirac semimetal state with no external perturbations, which has nothing to do with the interband optical scattering depicted by Fig.1(b). Recalling Fig.1, where the *direct* interband excitation generates the spin-flip and thus a triplet spin state for the exciton, we expect these processes could be important for realization of $p+ip$ excitonic insulator. We thus further present the study with taking into account a linearly-polarized light (photons) in the revised manuscript. We find that the photons have the renormalization effect on the TI surface by producing an effective p -wave interaction between electrons, whose form is the same as the interaction term in Eq.(10) of the manuscript. The underlying physical reason for this is nothing but the spin-flip mechanism during *direct* excitation (Fig.1(b)). According to the mean-field results above, it is found that the external field offers a promising approach to induce a $p+ip$ excitonic instability on TI surfaces.

Following the referee's comment, we have made significant improvements and fully discussed: the complete mean-field theory, the full phase diagram, the emergence of effective p -wave and s -wave interaction channels from the Hubbard model, and the stabilization of the $p+ip$ insulator by external field. We would like to thank the referee for these questions that motivate our further in-depth analysis. The revised manuscript now includes a complete study of the possible excitonic instabilities on TI surface as well as a promising experimental proposal to further stabilize the predicted $p+ip$ excitonic insulator with exotic properties, enriching our understandings on interacting topological orders.

In the following, we present some details for the additional calculations related to this question. Please also see the revised manuscript and the Methods.

1. *A complete mean-field theory on equal footing*—Let us consider the Dirac state $H_{Dirac} = \sum_{\mathbf{k}, \sigma, \sigma'} \psi_{\mathbf{k}, \sigma}^\dagger (\sigma_{\sigma, \sigma'})$.

$\mathbf{k} - \mu)\psi_{\mathbf{k}\sigma'}$ with the local Hubbard interaction between electrons,

$$H_I = \frac{V}{2} \sum_{\mathbf{k}, \mathbf{k}', \mathbf{q}} \sum_{\sigma} \psi_{\mathbf{k}+\mathbf{q}, \sigma}^{\dagger} \psi_{\mathbf{k}'-\mathbf{q}, \bar{\sigma}}^{\dagger} \psi_{\mathbf{k}', \bar{\sigma}} \psi_{\mathbf{k}, \sigma}. \quad (20)$$

Let us denote the four vectors respectively as $\mathbf{k}_1 = \mathbf{k}$, $\mathbf{k}_2 = \mathbf{k}'$, $\mathbf{k}_3 = \mathbf{k}' - \mathbf{q}$ and $\mathbf{k}_4 = \mathbf{k} + \mathbf{q}$. For zero temperature, only those degrees of freedom near μ become dominant. Due to conservation of momentum in 2D, only the forward scattering is least irrelevant for repulsive interactions in the low-energy window in the renormalization group sense, which consists two scattering channels (Rev. Mod. Phys. 66, 129 (1994)): $\mathbf{k}_1 = \mathbf{k}_3$ ($\mathbf{k}_2 = \mathbf{k}_4$) and $\mathbf{k}_1 = \mathbf{k}_4$ ($\mathbf{k}_2 = \mathbf{k}_3$).

Then we make a unitary transformation into the diagonal basis of H_0 with,

$$\psi_{\mathbf{k}, \uparrow} = \frac{1}{\sqrt{2}} e^{-i\theta} (c_{\mathbf{k}+} - c_{\mathbf{k}-}), \quad (21)$$

$$\psi_{\mathbf{k}, \downarrow} = \frac{1}{\sqrt{2}} (c_{\mathbf{k}+} + c_{\mathbf{k}-}). \quad (22)$$

For the scattering channel $\mathbf{k}_1 = \mathbf{k}_3$ ($\mathbf{k}_2 = \mathbf{k}_4$), after expansion, there are sixteen different combinations in total. The requirement of conservation of energy greatly simplifies the expression, leaving us six terms that read as: $(\hat{k}' \cdot \hat{k}) c_{k'+}^{\dagger} c_{k+}^{\dagger} c_{k'+} c_{k+}$, $-(\hat{k}' \cdot \hat{k}) c_{k'+}^{\dagger} c_{k-}^{\dagger} c_{k'+} c_{k-}$, $(\hat{k}' \cdot \hat{k}) c_{k'-}^{\dagger} c_{k+}^{\dagger} c_{k'-} c_{k+}$, $(\hat{k}' \cdot \hat{k}) c_{k'-}^{\dagger} c_{k-}^{\dagger} c_{k'-} c_{k-}$, $-(\hat{k}' \cdot \hat{k}) c_{k'+}^{\dagger} c_{k+}^{\dagger} c_{k'-} c_{k+}$, $(\hat{k}' \cdot \hat{k}) c_{k'+}^{\dagger} c_{k-}^{\dagger} c_{k'+} c_{k-}$. Only those scattering terms that involves two bands can open a gap on the Dirac cone, while the terms that within one band only renormalizes the Fermi velocity but do not contribute to any instability. Moreover, the translation invariant mean-field treatment to the second and fifth term contributes to a self-energy that renormalizes the chemical potential. With these consideration, the forward scattering channel $\mathbf{k}_1 = \mathbf{k}_3$ ($\mathbf{k}_2 = \mathbf{k}_4$) produces the following reduced interaction,

$$H_{int}^1 = -\frac{V}{4} \sum_{\mathbf{k}, \mathbf{k}', n} (\hat{k} \cdot \hat{k}') c_{\mathbf{k}, n}^{\dagger} c_{\mathbf{k}, -n} c_{\mathbf{k}', -n}^{\dagger} c_{\mathbf{k}', n}. \quad (23)$$

For the scattering channel $\mathbf{k}_1 = \mathbf{k}_4$ ($\mathbf{k}_2 = \mathbf{k}_3$), the same approach above leads to the following reduced interaction,

$$H_{int}^2 = -\frac{V}{4} \sum_{\mathbf{k}, \mathbf{k}', n} c_{\mathbf{k}, n}^{\dagger} c_{\mathbf{k}, -n} c_{\mathbf{k}', -n}^{\dagger} c_{\mathbf{k}', n}. \quad (24)$$

Both H_{int}^1 and H_{int}^2 , p-wave and s-wave type respectively, are reduced from the original Hubbard interaction, which are the least irrelevant channels that can gap out the Dirac cone. Now we treat both channels on equal footing and consider the most general case with two independent interaction strength V_p and V_s for the p-wave and s-wave channel respectively. Introducing the bosonic mean-fields $\Delta_s = V_s \sum \langle c_{\mathbf{k}-}^{\dagger} c_{\mathbf{k}+} \rangle$ and $\Delta_p = V_p \sum \langle \hat{k} c_{\mathbf{k}-}^{\dagger} c_{\mathbf{k}+} \rangle$, the total mean-field Hamiltonian reads as,

$$H_{tot} = \sum_{\mathbf{k}} C_{\mathbf{k}}^{\dagger} [v_F k \tau^z - \Delta_s - \Delta_p \cdot \hat{k} \tau^+ - \Delta_s^* - \Delta_p^* \cdot \hat{k} \tau^+] C_{\mathbf{k}} + \frac{|\Delta_s|^2}{V_s} + \frac{|\Delta_p|^2}{V_p}, \quad (25)$$

from which one can arrive at the self-consistent equations by minimizing the ground state energy. The zero temperature phase diagram is shown in the above figure.

For the local Hubbard model where $V_s = V_p$, we find from above result that the system lies in the s-wave region. Then, we discuss the interband optical processes illustrated by Fig.1, and how they can renormalize the Dirac surface state.

2. *The effect of interband optical excitations*– We have discussed in the manuscript that the interband optical excitations can generate excitons with spin triplet pairing and therefore favoring p -wave symmetry. Now we further quantify this idea by specifically consider a microscopic theory on the interacting TI surface with the application of a linearly-polarized laser with frequency $\bar{\omega}$ and light power I . In quantized language, the vector potential is formulated as

$$\mathbf{A}(\mathbf{r}) = \sum_{\mathbf{q}\lambda} \sqrt{\frac{2\pi}{\omega_{\mathbf{q}}}} \mathbf{e}_{\lambda} (a_{\mathbf{q}\lambda} + a_{-\mathbf{q}\lambda}^{\dagger}) e^{-i\mathbf{q}\cdot\mathbf{r}}, \quad (26)$$

where \mathbf{e}_{λ} denotes the polarization vectors. Then, the light is described in the quantized formalism as Hamiltonian of photons,

$$H_{ph} = \sum_{\mathbf{q}\lambda} \omega_{\mathbf{q}} (a_{\mathbf{q}\lambda}^{\dagger} a_{\mathbf{q}\lambda} + 1/2), \quad (27)$$

where $a_{\mathbf{q}\lambda}$ is the annihilation operator of photons. Two values $\lambda = 1, 2$ are needed due to the Coulomb gauge. Then we consider the minimal coupling between surface electrons and the photons,

$$H_{el} = \int d\mathbf{r} \psi_{\mathbf{r}}^{\dagger} \boldsymbol{\sigma} \cdot (-iv_F \nabla_{\mathbf{r}} - e\mathbf{A}(\mathbf{r})) \psi_{\mathbf{r}}, \quad (28)$$

where $\psi_{\mathbf{r}} = [\psi_{\mathbf{r}\uparrow}, \psi_{\mathbf{r}\downarrow}]^T$ is a two-component spinor, and $\psi_{\mathbf{r}\sigma}$ annihilates an electron with spin σ at position \mathbf{r} . v_F is the Fermi velocity of Dirac fermions. Inserting the quantized formula of $\mathbf{A}(\mathbf{r})$ into Eq.(28), we obtain in momentum space:

$$H_{el-ph} = -ev_F \sum_{\mathbf{k}\mathbf{q}\lambda\sigma\sigma'} \sqrt{\frac{2\pi}{\omega_{\mathbf{q}}}} (\boldsymbol{\sigma}_{\sigma\sigma'} \cdot \mathbf{e}_{\lambda}) (\psi_{\mathbf{k}\sigma}^{\dagger} a_{\mathbf{q}\lambda} \psi_{\mathbf{k}-\mathbf{q}\sigma'} + \psi_{\mathbf{k}\sigma}^{\dagger} a_{-\mathbf{q}\lambda}^{\dagger} \psi_{\mathbf{k}-\mathbf{q}\sigma'}). \quad (29)$$

For the laser beam being perpendicular to the surface, \mathbf{q} in Eq.(26) is along \hat{z} and the polarization in the x-y plane. Moreover, since $c \gg v_F$, the conservation of energy requires $|\mathbf{k}| \gg q$, therefore, $\mathbf{k} - \mathbf{q} \simeq \mathbf{k}$. With this bearing in mind, we integrate out the photons which are Gaussian and can be done exactly. This leads to an effective interaction described by the action,

$$S = - \int dt \sum_{\mathbf{k}\mathbf{k}'\mathbf{q}} \sum_{\alpha\beta\rho\gamma} \frac{2\pi e^2 v_F^2}{\omega_{\mathbf{q}}} \sum_{\lambda} (\boldsymbol{\sigma}_{\alpha\beta} \cdot \mathbf{e}_{\lambda}) (\boldsymbol{\sigma}_{\rho\gamma} \cdot \mathbf{e}_{\lambda}) \frac{1}{i\partial_t - \omega_{\mathbf{q}}} \psi_{\mathbf{k}\alpha}^{\dagger} \psi_{\mathbf{k}\beta} \psi_{\mathbf{k}'\rho}^{\dagger} \psi_{\mathbf{k}'\gamma}. \quad (30)$$

Since $\mathbf{A}(\mathbf{r})$ is polarized in the surface plane, $\mathbf{e}_1 = \hat{x}$ and $\mathbf{e}_2 = \hat{y}$, and therefore,

$$(\boldsymbol{\sigma}_{\alpha\beta} \cdot \mathbf{e}_{\lambda}) (\boldsymbol{\sigma}_{\rho\gamma} \cdot \mathbf{e}_{\lambda}) = \sigma_{\alpha\beta}^x \sigma_{\rho\gamma}^x + \sigma_{\alpha\beta}^y \sigma_{\rho\gamma}^y. \quad (31)$$

Inserting the above identity, Eq.(30) becomes,

$$S_{int} = - \int dt \sum_{\mathbf{k}\mathbf{k}'\mathbf{q}} \frac{2\pi e^2 v_F^2}{\omega_{\mathbf{q}}} \frac{1}{i\partial_t - \omega_{\mathbf{q}}} [(\psi_{\mathbf{k}\uparrow}^{\dagger} \psi_{\mathbf{k}\downarrow} + \psi_{\mathbf{k}\downarrow}^{\dagger} \psi_{\mathbf{k}\uparrow}) (\psi_{\mathbf{k}'\uparrow}^{\dagger} \psi_{\mathbf{k}'\downarrow} + \psi_{\mathbf{k}'\downarrow}^{\dagger} \psi_{\mathbf{k}'\uparrow}) + (-i\psi_{\mathbf{k}\uparrow}^{\dagger} \psi_{\mathbf{k}\downarrow} + i\psi_{\mathbf{k}\downarrow}^{\dagger} \psi_{\mathbf{k}\uparrow}) (-i\psi_{\mathbf{k}'\uparrow}^{\dagger} \psi_{\mathbf{k}'\downarrow} + i\psi_{\mathbf{k}'\downarrow}^{\dagger} \psi_{\mathbf{k}'\uparrow})]. \quad (32)$$

which is further reduced to

$$S_{int} = - \int d\omega d\omega' d\nu \sum_{\mathbf{k}\mathbf{k}'\mathbf{q}\sigma} \frac{4\pi e^2 v_F^2 \hbar^2}{\omega_{\mathbf{q}} (\nu - \omega_{\mathbf{q}})} \psi_{\mathbf{k}\sigma}^{\dagger}(\omega) \psi_{\mathbf{k}\bar{\sigma}}(\omega + \nu) \psi_{\mathbf{k}'\bar{\sigma}}^{\dagger}(\omega') \psi_{\mathbf{k}'\sigma}(\omega' - \nu). \quad (33)$$

Note that all \mathbf{q} vectors satisfy $\mathbf{q} = q\hat{z} = \bar{\omega}\hat{z}/c$ for a laser beam with certain frequency $\bar{\omega}$. The above action consists of an equal-time action that produces the following effective interaction between electrons for $\nu \ll \bar{\omega}$, i.e.,

$$H_{int}^{eff} = -\frac{4\pi e^2 v_F^2 n_{ph}}{\bar{\omega}^2} \sum_{\mathbf{k}\mathbf{k}'\sigma} \psi_{\mathbf{k}\sigma}^\dagger \psi_{\mathbf{k}\bar{\sigma}} \psi_{\mathbf{k}'\bar{\sigma}}^\dagger \psi_{\mathbf{k}'\sigma}, \quad (34)$$

where n_{ph} is the total number density of photons interacting with the electrons, which is proportional to the light power I . Physically, the first two operator $\psi_{\mathbf{k}\sigma}^\dagger \psi_{\mathbf{k}\bar{\sigma}}$ arises from the optical interband absorption process depicted by Fig.1 in the manuscript. We note that the first two operators in Eq.(34), $\psi_{\mathbf{k}\sigma}^\dagger \psi_{\mathbf{k}\bar{\sigma}}$, have the opposite spin and the same momentum \mathbf{k} . The opposite spin is due to the σ^x and σ^y Pauli matrix in Eq.(31), i.e., the interband optical absorption process is accompanied by the spin flip of electrons. The same momentum is due to the fact that $q \ll |\mathbf{k}|$ so that the photons favor direct interband excitation. Similarly, the second two operators in Eq.(34), $\psi_{\mathbf{k}'\bar{\sigma}}^\dagger \psi_{\mathbf{k}'\sigma}$ are originated from the interband optical emission process (also accompanied with spin flip). Therefore, the effective interaction Eq.(34) is a second order correction from the electron-photon coupling. Interestingly, H_{int}^{eff} has exactly the same form as the local Hubbard interaction in the scattering channel $\mathbf{k}_1 = \mathbf{k}_3$ (see above). After a unitary transformation to the band basis, we obtain,

$$H_{int}^{eff} = -\tilde{V} \sum_{\mathbf{k}\mathbf{k}'} \hat{k} \cdot \hat{k}' c_{\mathbf{k}+}^\dagger c_{\mathbf{k}-} c_{\mathbf{k}'-}^\dagger c_{\mathbf{k}'+}, \quad (35)$$

where the interaction strength, $\tilde{V} = \frac{4\pi e^2 v_F^2 n_{ph}}{\bar{\omega}^2}$, can be tuned by varying light frequency $\bar{\omega}$ or the light power I . According to results in the last section, one can tune H_{int}^{eff} through the external laser, so that the ground state can locate in the $p + ip$ -wave excitonic insulator region. This verifies the spin-flip mechanism and the formation of spin-triplet excitons demonstrated by Fig.1 in the manuscript.

4. “The technics in the paper are standard. All of them can be found in the previous literatures about the excitonic condensation. The paper is written in oral English in many sentences.”

Our response: We have now improved the English of the paper however we disagree with the Referee that all of the new papers published in Nature Communications need to invent new theoretical techniques. Our calculations might use standard field theory approaches, however they lead to novel physics including topological excitonic condensation with time reversal symmetry breaking and parity anomaly. These results are of significant interest to the community including both theorists and experimentalists in the field. Moreover, as mentioned above, evidences of our theoretical prediction, i.e., the excitonic insulators on TI surface, have been experimentally seen in a subsequent but independent paper (arXiv: 1810.10653 (2018)). We therefore believe the current work can stimulate more efforts in excitonic physics in topological insulators in the near future.

Summary of the changes

- 1) . In page 1, left column, the third and fourth paragraph, we added more discussion to further explain Fig.1.
- 2) . We modified the caption of figure 1, to illustrate the optical absorption process more clearly.
- 3) . In page 2, left column, right above the section “Model”, we added several sentences on the linearly-polarized light.
- 4) . In page 3, right column, below Fig.3, we added more discussion on the quantity β .
- 5) . In section “Excitonic insulator on the TI surface”, the first paragraph, we added more discussion to emphasize the difference between our proposal and earlier works.
- 6) . We added new calculations and a new Fig.4(c) on the phase diagram with treating the s -wave and p -wave channel on equal footing. We also modified the figure caption of Fig.4 to include the new results.
- 7) . In section “Excitonic insulator on the TI surface”, the second paragraph, we added more discussion on the wrapping effect, and how our results are robust on it.
- 8) . In section “Excitonic insulator on the TI surface”, we added much new content on our addition calculations with considering the s -wave and p -wave instability on equal footing.
- 9) . In section “Excitonic insulator on the TI surface”, we added a whole new paragraph to discuss a realistic experimental way to drive the system into the $p + ip$ excitonic insulator, i.e., by the application of a linearly-polarized perpendicular light to the TI surface.
- 10) . In the second paragraph of the section “Discussion”, we added much more discussion on (1) the long-range Coulomb interaction (2) finite temperature case (3) thermal fluctuation around the mean-field solution (4) the experimental advantage or uniqueness of our proposal.
- 11) . In Methods, we added a new section to explain how to derive Eq.(10) from Eq.(9).
- 12) . In Methods, we added a new section to show details on the renormalization effect due to the linearly-polarized light starting from a microscopic model based on electron-photon couplings.
- 13) . We added ten more references. Some of them are suggested by the referees, others can help to explain our new content well.
- 14) . We added a note to discuss a recent experiment, where a laser-induced non-equilibrium version of our proposal has been observed in Bi_2Se_3 .

REVIEWERS' COMMENTS:

Reviewer #1 (Remarks to the Author):

The authors have answered all my questions in details and I recommend the publication.

Reviewer #3 (Remarks to the Author):

The authors have fully addressed my comments and questions. They also performed careful revisions to the manuscript, making their discussions more complete. Therefore, I recommend publication of this manuscript in Nature Communications.

Reviewer #4 (Remarks to the Author):

In the revised draft, the author proposed the interband optical pumping to achieve the $p+ip$ excitonic insulator. I think it makes sense. But, I think the presentation in the abstract should be changed. The author should weaken or delete this sentence "Due to the spin-momentum locking, the electron-electron coupling intrinsically generates a p -wave interaction component that results in a significant parameter region that stabilizes the $p + ip$ excitonic insulator." , but emphasize this sentence "We also demonstrate that the interband optical processes and additional photons can renormalize the surface electrons, driving the system towards the proposed $p + ip$ instability." Because the optical pumping is the primary and the interaction is the secondary to get the ground state with $p+ip$ symmetry.

In general, the ground states with chiral symmetry are degenerate. i.e., the states with $p+ip$ and $p-ip$ symmetry. The pristine chiral states should involve the two regimes with $p+ip$ and $p-ip$, respectively. What is the situation in the present study? Can the author give some discussions with the experiments.

After the above questions are addressed, I could recommend the revised draft for Nature Communications.

Response to the Reviewers

We thank the Referees for reading our manuscript and the recommendation of publication on Nature Communications. Here are our brief responses to the points and suggestions they have raised.

Reply to referee 4

1. In the revised draft, the author proposed the interband optical pumping to achieve the p+ip excitonic insulator. I think it makes sense. But, I think the presentation in the abstract should be changed. The author should weaken or delete this sentence “Due to the spin-momentum locking, the electron-electron coupling intrinsically generates a p-wave interaction component that results in a significant parameter region that stabilizes the p + ip excitonic insulator.” , but emphasize this sentence “We also demonstrate that the interband optical processes and additional photons can renormalize the surface electrons, driving the system towards the proposed p + ip instability.” Because the optical pumping is the primary and the interaction is the secondary to get the ground state with p+ip symmetry.

Response: We understand the referee’s concern and agree with this suggestion. We have deleted the sentence “Due to the spin-momentum locking, the electron-electron coupling intrinsically generates a p-wave interaction component that results in a significant parameter region that stabilizes the p + ip excitonic insulator”, and have emphasized the effect of interband optical pumping in the abstract. This modification indeed makes the main findings better demonstrated and more clear to readers, as suggested by the referee.

2. “In general, the ground states with chiral symmetry are degenerate, i.e., the states with p+ip and p-ip symmetry. The pristine chiral states should involve the two regimes with p+ip and p-ip, respectively. What is the situation in the present study? Can the author give some discussions with the experiments.”

Response: The referee is correct on this point. It is clear from the mean-field study in our manuscript that the exciton order parameters with $p+ip$ and $p-ip$ symmetry are energetically degenerate with each other. The situation here is the same as that in the pristine chiral p -wave superconductors. The twofold degenerate states with $p+ip$ and $p-ip$ symmetry, which enjoy the opposite chiralities and therefore the opposite Chiral Chern numbers ($C_c = 1/2$ and $C_c = -1/2$ respectively), can be spontaneously formed in different regions in the system via symmetry breaking, generating domain walls featuring the change of chiral Chern number $\Delta C_c = 1$. The static properties of the domain walls and their dynamics are interesting topics for further studies. We thank the referee for the illuminating question. We have also added some brief discussions on the experiments related to our work.